# A Survey of Automatic Hallucination Evaluation on Natural Language Generation

## Abstract

The proliferation of Large Language Models (LLMs) has introduced a critical challenge: accurate hallucination evaluation that ensures model reliability. While Automatic Hallucination Evaluation (AHE) has emerged as essential, the field suffers from methodological fragmentation, hindering both theoretical understanding and practical advancement. This survey addresses this critical gap through a comprehensive analysis of 105 evaluation methods, revealing that 77.1% specifically target LLMs, a paradigm shift that demands new evaluation frameworks. We formulate a structured framework to organize the field, based on a comprehensive survey of foundational datasets and benchmarks and a taxonomy of evaluation methodologies, which together systematically document the evolution from pre-LLM to post-LLM approaches. Beyond taxonomical organization, we identify fundamental limitations in current approaches and their implications for real-world deployment. To guide future research, we delineate key challenges and propose strategic directions, including enhanced interpretability mechanisms and integration of application-specific evaluation criteria, ultimately providing a roadmap for developing more robust and practical hallucination evaluation systems.

## 1 Introduction

Hallucination in Natural Language Generation (NLG) typically refers to situations where generated text contradicts or lacks support from source input or external knowledge. While the term *hallucination* is relatively recent, the underlying challenges of ensuring factual consistency and faithfulness have been a long-standing concern in the field. Early rule-based and template-based NLG systems, for instance, were designed to prioritize factual accuracy, often at the cost of linguistic fluency (Deemter, 2024; Ji et al., 2023; Gatt & Krahmer, 2018). As text generation models evolved, technologies like Large Language Models (LLMs) achieved grammatical correctness and fluency nearly indistinguishable from human writing (Dou et al., 2022; Brown et al., 2020). Consequently, hallucination has emerged as a prominent concern demanding urgent attention. Automatic hallucination evaluation will be crucial for advancing LLMs toward greater reliability and safety. This paper presents a comprehensive survey of Automatic Hallucination Evaluation (AHE) methods, documenting current advances in hallucination detection while identifying future research directions.

The concept of hallucination initially described grammatically correct but semantically inaccurate content relative to source input (Lee et al., 2018). This phenomenon appeared commonly in tasks like Summarization (See et al., 2017; Maynez et al., 2020), Neural Machine Translation (NMT) (Raunak et al., 2021), and Data-to-Text Generation (Lin et al., 2024; Rebuffel et al., 2021; Thomson & Reiter, 2021), where source information remained well-defined. The paradigm shifted dramatically with LLMs like ChatGPT (OpenAI, 2022). A wide array of NLG tasks, from dialogue systems and creative writing to complex question answering, became achievable through prompting LLMs with designed instructions (Ouyang et al., 2022). However, their responses frequently contain hallucinations deviating from input or established world knowledge (Jesson et al., 2024), presenting significant evaluation challenges.

To clarify these challenges, we distinguish between faithfulness and factuality, two closely related yet distinct concepts. Faithfulness measures output consistency with given source input, while factuality assesses alignment

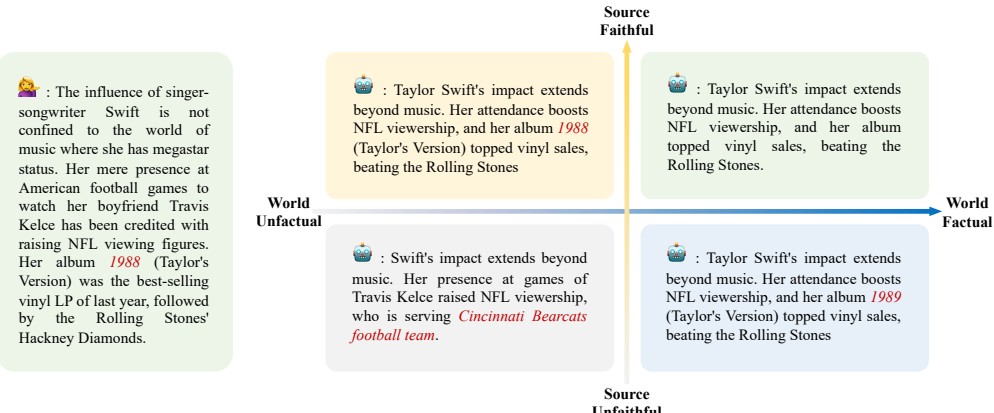

Figure 1: Source Faithful Error (SFE) and World Factual Error (WFE) examples. The correct album is *"1989"*, but the source document contains incorrect information. If the generated text says *"1988"*, it is SF but has WFE. If it corrects to *"1989"*, it is WF but has SFE. When the text exhibits both SFE and WFE, it often includes non-factual content not from the source, e.g. the incorrect statements about *Travis Kelce* ==is serving the Cincinnati Bearcats football team==. Otherwise, if no such errors are present, the text should be both SF and WF.

with established real-world knowledge. Despite frequent usage, these terms often become conflated (Mishra et al., 2024b; Huang et al., 2023a; Dong et al., 2022; Xie et al., 2021), creating evaluation ambiguity. This paper provides clearer distinctions by introducing precise terminology: Source Faithfulness (SF) and World Factuality (WF). SF measures how accurately generated output reflects source input consistency. SF operates within limited scope, as specific sources can substantiate generated text. WF assesses whether the generated output aligns with general world knowledge and facts. WF presents more expansive challenges, extending beyond specific sources to consider broader common sense and established knowledge, which proves difficult to collect and encode comprehensively (Gupta et al., 2024; Garrido et al., 2024). Recent studies increasingly recognize the critical importance of measuring SF and WF in generated text.

Evaluating SF versus WF aspects requires different source information, closely tied to specific tasks. In NMT, translations detached from source text are deemed unfaithful (Dale et al., 2023a). In summarization, summaries should maintain source document faithfulness, though some hallucinations may still be factually correct with respect to external facts (Dong et al., 2022). In LLM-based tasks, hallucinations exhibit greater diversity ==compared to earlier, task-specific models,== often encompassing both SF and WF issues simultaneously. LLMs face unique challenges, including outdated world information and false-premise questions (Kasai et al., 2023; Yuan et al., 2024). Figure 1 illustrates these error types through a four-quadrant framework. In light of these task-specific differences and evolving error types, we define the scope of this survey to clearly delimit the boundaries of our analysis and maintain conceptual clarity.

## 1.1 Scope of the Survey

We followed the PRISMA framework to conduct a systematic search for studies on hallucination evaluation in NLG. The search was performed across multiple sources covering the period from January 2020 to July 2025. Specifically, we queried DBLP[1] as our primary structured database and complemented it with open repositories and websites, including ACL Anthology[2] and Google Scholar[3]. In addition, we manually checked the proceedings of other major AI/ML conferences (ICLR, ICML, NeurIPS, and AAAI)[4] to ensure coverage of the most recent state-of-the-art work. We constructed search queries by combining the first set of terms:

---

[1] https://dblp.org/

[2] https://aclanthology.org/

[3] https://scholar.google.com/

[4] International Conference on Learning Representations (ICLR), International Conference on Machine Learning (ICML), Conference on Neural Information Processing Systems (NeurIPS), The Association for the Advancement of Artificial Intelligence (AAAI)

*hallucination, factuality, faithfulness* with the second set: *evaluation, assessment, measurement, benchmark, dataset*, in titles and metadata. Duplicates across sources were removed before screening. The selection process involved title and abstract screening followed by full-text eligibility checks. We exclude hallucination mitigation techniques, purely human evaluation methods, and multimodal systems to maintain a focused scope on text-only automated evaluation. A PRISMA flow diagram summarizes the number of records retrieved, screened, and included in Appendix A.

This survey systematically organizes AHE methods across datasets and methodologies. Our goal is to provide a comprehensive account of how hallucination has been assessed across different eras of models, specifically contrasting the pre-LLM era, marked by smaller, task-specific systems, with the post-LLM era, characterized by powerful, instruction-tuned models with broader generative capabilities and increased unpredictability. We analyze and compare methods from both periods through a central framework that distinguishes between the SF and WF perspectives, which shape the definition, detection, and measurement of hallucination across evaluation techniques.

## 1.2 Compare with Existing Surveys

Several surveys have touched upon methods for evaluating hallucinations in LLMs, though often only briefly or without detailed analysis (Huang et al., 2023b; Zhang et al., 2023c; Ji et al., 2023; Huang et al., 2021). These surveys primarily focus on either pre-LLM or early-stage LLM techniques and do not cover more recent developments in the field. Consequently, they do not provide a comprehensive categorization of benchmarks or systematically summarized evaluator processes. Furthermore, they lack a comparative analysis of methods across different stages, leading to an absence of in-depth analysis regarding their details, strengths, and weaknesses. In contrast, our survey presents a unified and up-to-date (to July 2025) overview of AHE methodologies, structured around a framework that categorizes evaluation methods, as illustrated in Figure 2

## 1.3 Structure of the Survey

This survey is structured around the foundational components of AHE research and a taxonomy of evaluation methodologies. We begin with **datasets and benchmarks** (§2) as the essential foundation, focusing on data availability and diversity across various tasks. Following this, this survey organizes AHE methodologies into three core paradigms (§3): (1) **Reference-based Evaluation**, which involves collecting evidence and comparing it to the generated text; (2) **Reference-free Evaluation**, which analyzes the model's internal states or consistency without external grounding; and (3) **LLM-based Evaluation**, which leverages other powerful language models as the primary evaluation tool. Finally, we synthesize the findings of our survey in a critical **discussion** (§4) that explores the field's primary challenges, such as the evolving nature of hallucinations and the limitations of current approaches. Building on this analysis, we conclude by outlining promising future directions for research. We also present Table 4, Table 5, and Table 6 for all the methods surveyed in this paper, including key aspects discussed in the following sections. This structure is designed to guide the reader from the foundational components of AHE to the current research frontier of trustworthy NLG systems.

## 2 Foundations of AHE: Datasets and Benchmarks

This section introduces datasets and benchmarks developed for evaluating model hallucination. Of the evaluators surveyed (Appendix C.1), 46.7% present their datasets or benchmarks for evaluation. The evolution has shifted from task-specific methods to general factuality assessments, with recent works focusing on more practical and diverse domains, adapting design patterns to various usage scenarios.

## 2.1 Task-specific Benchmarks

Although common task-specific datasets are not originally curated with hallucination detection in mind, they often contain instances of hallucinated content as a byproduct of the task, making them valuable resources for hallucination evaluation. In particular, the summarization task has seen substantial efforts in this regard,

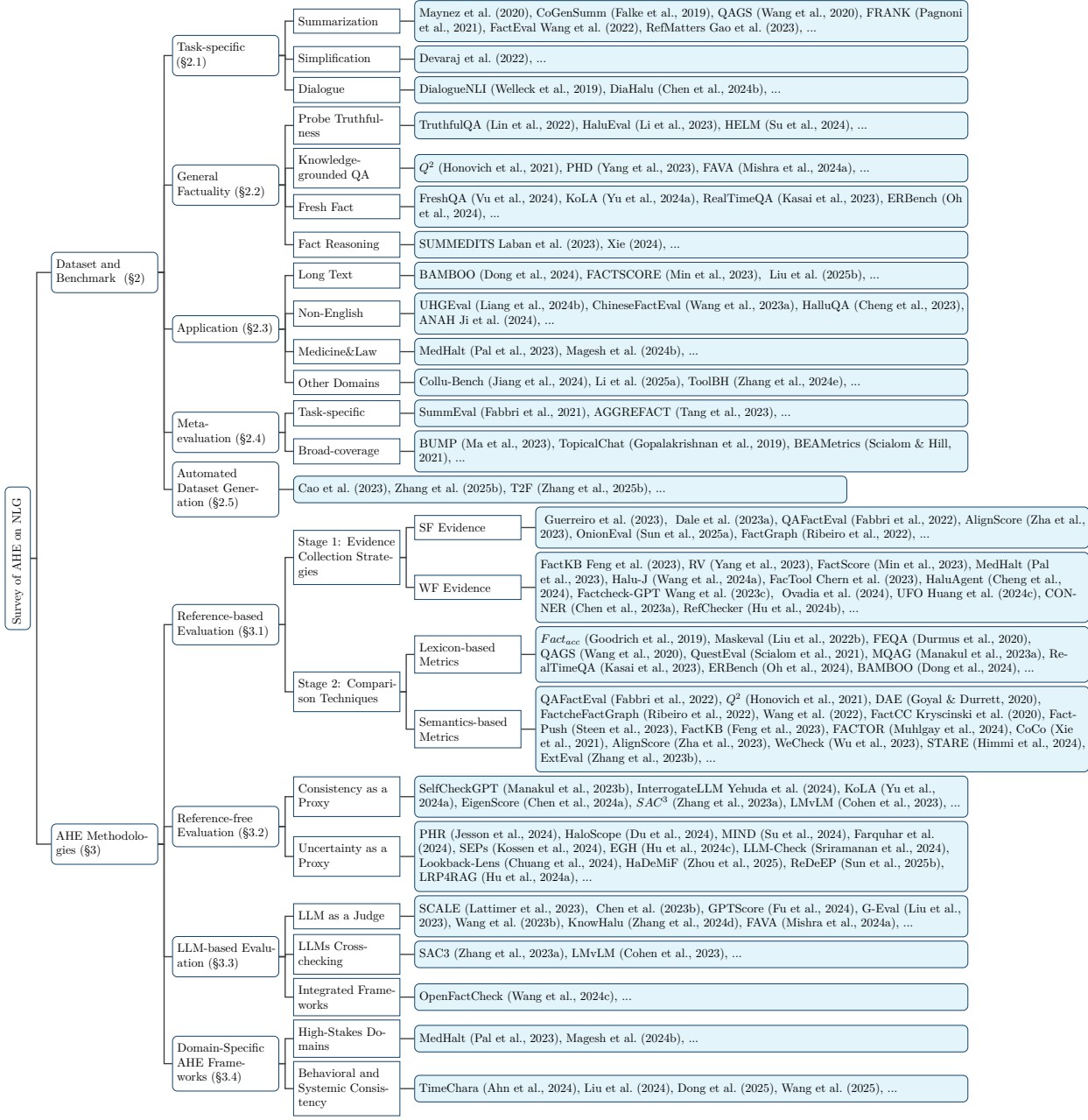

Figure 2: Taxonomy of AHE methods (highlighted nodes with shading) based on the distinct techniques employed at each stage of the pipeline.

where many studies have manually assessed model-generated summaries and released annotated datasets to facilitate research. Early annotation efforts on popular news summarization datasets like XSum and CNN/DM primarily used binary labels to assess hallucination. Some of these focused exclusively on SF, where a summary is checked against its source document (Falke et al., 2019; Wang et al., 2020). Others adopted a broader scope, providing annotations for both SF and WF (Maynez et al., 2020; Huang et al., 2020).

While early benchmarks could identify the presence of hallucinations, their binary labels offered limited insight into the specific nature of these errors, such as their type, severity, or contextual impact. This led to a second wave of benchmarks with more sophisticated, fine-grained typologies of hallucinations.

FRANK (Pagnoni et al., 2021) was a pivotal benchmark in this direction. This trend extended to dialogue summarization with datasets like FactEval (Wang et al., 2022) and RefMatters (Gao et al., 2023), which introduced detailed error categorization. More recently, benchmarks have begun to focus not just on identifying errors, but on understanding their origins. For instance, FaithBench (Bao et al., 2024) isolates challenging summaries that fool state-of-the-art detectors, while SummaCoz (Luo et al., 2024) provides explanations for why hallucinations occur, enabling deeper causal analysis.

Beyond summarization, Devaraj et al. (2022) propose a taxonomy of factual errors, namely, information insertion, deletion, and substitution, in the context of the text simplification task, using data from the Newsela (Xu et al., 2015) and Wikilarge (Zhang & Lapata, 2017) datasets. In the domain of dialogue generation, factual consistency has also received growing attention. DialogueNLI (Welleck et al., 2019) provides sentence-level entailment labels to assess the logical consistency between utterances. Going beyond sentence-level evaluation, DiaHalu (Chen et al., 2024b) introduces a comprehensive benchmark at the dialogue level, incorporating both SF and WF annotations. Expanding to other generation settings, RAGTruth (Niu et al., 2024) addresses hallucination in Retrieval-Augmented Generation (RAG) systems. It offers fine-grained annotations that distinguish between evident and subtle hallucinations, thereby supporting more robust and nuanced evaluation in retrieval-based contexts, covering summarization, QA, and data-to-text tasks.

As LLMs continue to advance, the boundaries between tasks are becoming increasingly blurred, indicating that future data development efforts should aim to support more general and cross-domain applications. Embodying this shift, benchmarks like HalluMix (Emery et al., 2025) have been proposed, offering a task-agnostic and multi-domain collection of real-world data designed to support the development of more universally applicable AHE methods.

## 2.2 General Factuality Benchmarks

To assess LLMs' overall capacity to avoid hallucinations, research has moved toward more generalized evaluation protocols. These benchmarks aim to probe hallucination tendencies in broader, open-ended scenarios, making them more reflective of real-world use cases. They often emphasize WF and are frequently structured around Question-Answering (QA) formats, assessing not only factual recall but also complex behaviors like reasoning and truthfulness.

**Knowledge-grounded QA**  A significant portion of these benchmarks evaluates factuality by grounding model responses in large-scale knowledge corpora, primarily Wikipedia. For instance, HaluEval (Li et al., 2023) and PHD (Yang et al., 2023) use this approach to detect hallucinations in model responses. To enable more detailed analysis, benchmarks like FAVA (Mishra et al., 2024a) offers more fine-grained annotations through tagged elements in the model-generated text. Pushing this granularity even further, HADES (Liu et al., 2022a) provides reference-free hallucination annotations at the token level. Others adopt a multi-choice QA format, such as FACTOR (Muhlgay et al., 2024), which uses fine-grained error types inspired by Pagnoni et al. (2021). Similarly, $Q^2$ (Honovich et al., 2021) offers an annotated dataset to check for consistency against provided knowledge in dialogue.

**Probing Model Truthfulness**  Beyond factual recall, a crucial line of research evaluates the broader behavior of truthfulness. TruthfulQA (Lin et al., 2022) was a landmark in this area, highlighting the tension between being informative and being truthful, and arguing that models should learn to hedge rather than invent answers, while benchmarks like SimpleQA (Wei et al., 2024a) focus on the scalable evaluation of factuality for more straightforward, short-form queries. This perspective was extended by HalluLens (Bang et al., 2025), which specifically assesses a model's ability to refuse answering questions about non-existent entities. Other benchmarks like THaMES (Liang et al., 2024a) jointly assess SF and WF by pairing hallucinated and correct answers to test a model's discernment. To facilitate deeper investigation into model behavior, frameworks like HELM (Su et al., 2024) even provide snapshots of models' internal states during generation.

**Fresh Facts**  As the world is constantly changing, a critical question arises: how can we assess whether LLMs possess up-to-date, dynamic knowledge? To address this, several benchmarks focus on constructing time-sensitive datasets. Some define categories of fresh events and build continuous collection workflows (Vu

et al., 2024; Kasai et al., 2023; Yu et al., 2024a). This principle is embodied by dynamic benchmarks like FactBench (Bayat et al., 2025), which continuously sources claims from "in-the-wild" data to avoid both knowledge staleness and training data contamination. Others integrate external tools like search engines (Zhang et al., 2024e) or structured databases (Oh et al., 2024) to support real-time applications. Findings from these investigations indicate that larger model sizes do not necessarily improve factuality. Instead, factors such as the quality of training data and the design of response strategies play critical roles in determining a model's ability to minimize hallucinations.

**Fact Reasoning**   Reasoning with LLMs in hallucination evaluation is challenging due to its multi-step nature. While benchmarks like SUMMEDITS provide structured protocols for assessing factual consistency across different domains (Laban et al., 2023), recent work argues that this does not fully capture errors embedded within the reasoning process itself. For instance, Xie (2024) demonstrate that the order of the reasoning steps is critical. They propose that even with correct facts, an illogical sequence can lead to a flawed conclusion, and thus, the reasoning order itself should be used as a benchmark. This highlights a crucial distinction: a comprehensive evaluation should assess not only the hallucination of each reasoning step, but also the logical coherence among them.

In summary, benchmarks for general factuality are constructed via two primary approaches: structured formats like multiple-choice questions and more fine-grained human annotation of open-ended outputs. The former is effective for probing specific knowledge and behavioral traits like appropriate refusal, while the latter excels at capturing nuanced, contextual hallucinations. Collectively, these efforts push AHE beyond task-specific confines, aiming for evaluations that reflect the broad and unpredictable nature of real-world applications.

## 2.3   Benchmarks for Application

As LLMs are deployed in increasingly complex and critical settings, recent benchmarks have focused on evaluating hallucinations across these challenging frontiers: long-form contexts or outputs, global multilingual scenarios, high-stakes specialized domains, and diverse interdisciplinary areas.

**Long Context/Generation**   Evaluating hallucinations in extended contexts remains a significant challenge, as long-form outputs often contain a complex mixture of factual and hallucinated information, making assessment highly nuanced (Liu et al., 2025b). To tackle this, benchmarks often decompose long texts into fine-grained factual units for precise evaluation. For example, FactScore (Min et al., 2023) evaluates long-form biographies by breaking the text into atomic facts and assigning each a binary label. Similarly, BAMBOO (Dong et al., 2024) integrates hallucination detection as a key task within its broader multi-task benchmark for long-context scenarios.

**Non-English Languages**   As LLMs become global technologies, evaluating hallucinations in non-English languages is crucial. Notable efforts have targeted Chinese, with benchmarks assessing local contexts and cultural nuances (Liang et al., 2024b; Wang et al., 2023a; Cheng et al., 2023), providing simple QA formats for factuality (He et al., 2025), and offering fine-grained, automatically constructed datasets (Zhang et al., 2025b). This focus extends to other languages like Korean, with specialized resources featuring multiple-choice answer designs (Seo & Lim, 2025). Beyond single-language resources, a diverse ecosystem of multilingual and cross-lingual benchmarks has emerged. Some provide broad, fine-grained coverage across many languages (Zhang et al., 2024c; Abdaljalil et al., 2025) or adapt established paradigms like fact verification for multilingual settings (Zhang et al., 2025a). Deeper grounding is achieved by leveraging external knowledge graphs (Lavrinovics et al., 2025) or internal semantic representations like Abstract Meaning Representation (Regan et al., 2024). Other benchmarks address specific cross-lingual challenges, such as disentangling hallucinations from translation infelicities (Dale et al., 2023b), prompting models for self-annotation in bilingual contexts (Ji et al., 2024), or focusing on critical applications like RAG (Jiang et al., 2025).

**High-Stakes Domains**   In specialized domains like medicine and law, hallucinations can carry severe real-world consequences, necessitating the development of highly specialized evaluation datasets. In the medical field, benchmarks have been created to systematically detect clinical hallucinations. These include

| Category | Dataset | Task | Size | Label Type | Links |
|---|---|---|---|---|---|
| **Task-specific** | DialogueNLI (Welleck et al., 2019) | Dialogue | 343k pairs | Entailment/contradiction/neutral | GitHub |
| | CoGenSumm (Falke et al., 2019) | Summarization | 100 articles | Sentence correct/incorrect | Dataset Link |
| | XSumFaith (Maynez et al., 2020) | Summarization | 500 articles | Span intrinsic/extrinsic hallucination | GitHub |
| | QAGS (Wang et al., 2020) | Summarization | 474 articles | Consistent/inconsistent | GitHub |
| | Polytope (Huang et al., 2020) | Summarization | 1.5k summaries | Intrinsic/extrinsic hallucination | GitHub |
| | FRANK (Pagnoni et al., 2021) | Summarization | 2.25k summaries | Relation/entity/circumstance/ coreference/discourse/out-of-article/ gramma errors | GitHub |
| | Falsesum (Utama et al., 2022) | Summarization | 2.97k articles | Consistent/inconsistent | GitHub |
| | FactEval (Wang et al., 2022) | Dialogue summarization | 150 dialogues | Consistent/inconsistent | GitHub |
| | Devaraj et al. (2022) | Text simplification | 1.56k pairs | Insertion/deletion/substitution | GitHub |
| | NonFactS (Soleimani et al., 2023) | Augmented summarization | 400k samples | Non-factual summaries | GitHub |
| | RefMatters (Gao et al., 2023) | Dialogue summarization | 4k pairs | FRANK errors | GitHub |
| | DiaHalu (Chen et al., 2024b) | Dialogue generation | 1.0k samples | Dialogue-level factuality/ faithfulness | GitHub |
| | TofuEval (Tang et al., 2024) | Dialogue summarization | 1.5k pairs | Consistent/inconsistent | GitHub |
| | RAGTruth (Niu et al., 2024) | RAG systems | 2.97k samples | Evident/subtle conflict/baseless | GitHub |
| | SummaCoz (Luo et al., 2024) | Summarization | 6.07k summaries | Explanation | HF Dataset |
| | FaithBench (Bao et al., 2024) | Summarization | 750 samples | Questionable/benign/unwanted | GitHub |
| **General Factuality** | Q2 (Honovich et al., 2021) | Knowledge-based dialogue QA | 750 samples | Consistent/inconsistent | GitHub |
| | HADES (Liu et al., 2022a) | Free-form Generation | 34k instances | Token-level hallucination | GitHub |
| | TruthfulQA (Lin et al., 2022) | Truthfulness QA | 817 pairs | QA truthfulness | GitHub |
| | FACTOR (Muhlgay et al., 2024) | Multi-choice | 4.27k samples | FRANK errors | GitHub |
| | HaluEval (Li et al., 2023) | QA/Summarization/ dialog/general | 35K samples | Hallucinations yes/no | GitHub |
| | PHD (Yang et al., 2023) | Passage-level QA | 300 entities | factual/non-factual/unverifiable | GitHub |
| | FAVA (Mishra et al., 2024a) | General queries | 200 queries | Entity/relation/contradictory/ invented/subjective errors/ unverifiable | Project Page |
| | THaMES (Liang et al., 2024a) | General QA | 2.1k samples | Correct/hallucinated | GitHub |
| | HELM (Su et al., 2024) | LLM continue generation | 1.2k passages | Hallucination/non-hallucination | GitHub |
| | HalluLens (Bang et al., 2025) | LLM generation | 130k instances | Intrinsic/extrinsic hallucination / factuality | GitHub |
| | FreshLLMs (Vu et al., 2024) | Time-sensitive QA | 599(June,2025) pairs | Fast/slow/never changing/ false premise | GitHub |
| | ERBench (Oh et al., 2024) | Knowledge-based LLM QA | Not specified | Binary/multi-choice | GitHub |
| | KOLA (Yu et al., 2024a) | Knowledge-based LLM generation | 2.15k samples | Correct/incorrect | GitHub |
| | RealtimeQA (Kasai et al., 2023) | Real-time knowledge | 4.3k(June,2023) pairs | Correct/retrieval/ reading comprehension error | GitHub |
| | FactBench (Bayat et al., 2025) | Dynamic Factuality Eval | Continuously growing | Factually Correct/Incorrect | GitHub |
| | SimpleQA (Wei et al., 2024a) | Short Factuality QA | 2k prompts | Factual / Not Factual | HF Dataset |

Table 1: Overview of AHE datasets/benchmarks for task-specific and general factuality. "HF" indicates "HuggingFace".

structured, multi-facet fact-testing pipelines like MedHalt (Pal et al., 2023), datasets of synthetic, high-risk QA pairs built upon medical literature, such as MedHallu (Pandit et al., 2025), to varied validation processes such as manual expert verification (Joseph et al., 2024) and automated methods that decompose texts into atomic facts for scalable analysis (Seo et al., 2024). Similarly, in the legal domain, datasets have been compiled to cover complex legal questions across dimensions like jurisdiction, time-sensitivity, and false premises, enabling more rigorous evaluation of legal response generation (Magesh et al., 2024b).

| Category | Dataset | Task | Size | Label Type | Links |
|---|---|---|---|---|---|
| **Application** | FactScore (Min et al., 2023) | Long-form biography | 6.5k samples | Support/unsupport | GitHub |
| | BAMBOO (Dong et al., 2024) | Long-context | 1.5k samples | SenHallu, AbsHallu | GitHub |
| | ChineseFactEval (Wang et al., 2023a) | Chinese multi-domain | 125 prompts | Factual/non-factual | Project Page |
| | HalluQA (Cheng et al., 2023) | Chinese QA | 450 questions | Misleading/misleading-hard/knowledge | GitHub |
| | UHGEval (Liang et al., 2024b) | Chinese news | 5k samples | Hallucination/non-halluciantion | GitHub |
| | ANAH (Ji et al., 2024) | Chinese/English LLM generation | 4.3k generation | Contradictory/unverifiable/no fact | GitHub |
| | HalOmi (Dale et al., 2023c) | Multilingual translation | 18 langs × (144-197 pairs) | Hallucination, omission | GitHub |
| | Chinese SimpleQA (He et al., 2025) | Chinese Factuality QA | 10k questions | Correct/Incorrect/Refusal | GitHub |
| | C-FAITH (Zhang et al., 2025b) | Chinese Summarization/QA | 4k summaries | Span-level annotation | GitHub |
| | Bi'an (Jiang et al., 2025) | RAG (EN/ZH) | 5.2k triplets | Supported/Partially/Not Supported | GitHub |
| | HalluVerse25 (Abdaljalil et al., 2025) | Multilingual Q&A (25 langs) | 12.5k samples | Binary + fine-grained category | HF Dataset |
| | Poly-FEVER (Zhang et al., 2025a) | Multilingual Fact Verification | ~185k claims | Supported/Refuted/NotEnoughInfo | HF Dataset |
| | MASSIVE (Regan et al., 2024) | Multilingual AMR (51 langs) | 1M utterances | Semantic fidelity (Smatch) | GitHub |
| | MultiHal (Lavrinovics et al., 2025) | KG-grounded QA (8 langs) | 4.8k questions | Consistent/Inconsistent with KG | HF Dataset |
| | K-HALU (Seo & Lim, 2025) | Korean QA | 3.5k questions | Correct/Hallucinated | GitHub |
| | MedHalt (Pal et al., 2023) | Medical tests | 25.64k samples | Groundedness/hallucination | Project Page |
| | MedHallu (Pandit et al., 2025) | Medical QA | 10k samples | Hard/medium/easy hallucination | Project Page |
| | LegalHallu (Magesh et al., 2024b) | Legal QA | 745k samples | Correctness/groundedness | HF Dataset |
| | SUMMEDITS (Laban et al., 2023) | Multi-domain | 6.35k samples | Consistent/inconsistent | HF Dataset |
| | DefAn (Rahman et al., 2024) | Cross-domain Q&A | 3k questions | Factual/Hallucinated | GitHub |
| | HalluMix (Emery et al., 2025) | Multi-domain Detection | 7.7k examples | Binary (Hallucination/Faithful) | GitHub |
| | ToolBeHonest (Zhang et al., 2024e) | Tool-augmented LLM | 700 samples | missing necessary tools/potential tools/limited functionality tools | GitHub |
| | RoleBench (Kong et al., 2024) | Role-Playing Agents | 2k instances | In-Character/Out-of-Character | HF Dataset |
| | Molecular Mirage (Li et al., 2025a) | Molecular QA | 1.1k questions | Binary (hallucination/faithful) | GitHub |
| | Collu-Bench (Jiang et al., 2024) | Code Generation | 1.2k prompts | Boolean (likely to hallucinate) | GitHub |
| | TIB (Li et al., 2025b) | Traffic Incident QA | 2.5k pairs | Multi-class (No/Mild/Severe) | Paper Link |
| **Meta-evaluation** | Wizard of Wikipedia (Dinan et al., 2019) | Knowledge-based dialogue eval | 22.3k dialogues | Knowledge selection, response generation | Project Page |
| | TopicalChat (Gopalakrishnan et al., 2019) | Knowledge-based dialogue eval | 10.79k dialogues | Knowledge source | GitHub |
| | SummEval (Fabbri et al., 2021) | Summarization metric eval | 1.6k summaries | Consistent/inconsistent | GitHub |
| | BEAMetrics (Scialom & Hill, 2021) | Multi-task metric eval | Not specified | Coherence | GitHub |
| | CI-ToD (Qin et al., 2021) | Task-oriented dialogue | 3.19k dialogues | Consistent/inconsistent | GitHub |
| | SummaC (Laban et al., 2022) | Summarization metric eval | Not specified | Consistent/inconsistent | GitHub |
| | BEGIN (Dziri et al., 2022b) | Knowledge-based dialogue | 12k turns | Fully/not attributable/generic | GitHub |
| | FaithDial (Dziri et al., 2022a) | Dialogue eval | 5.65k ddialogues | BEGIN, VRM | HF Dataset |
| | DialSumMeval (Gao & Wan, 2022) | Dialogue summarization metric eval | 1.5k summaries | Consistent/inconsistent | GitHub |
| | TRUE (Honovich et al., 2022) | Cross-task metric eval | ~200k samples | Consistent/inconsistent | GitHub |
| | AGGREFACT (Tang et al., 2023) | Summarization metric eval | 59.7k samples | Consistent/inconsistent | HF Dataset |
| | FELM (Chen et al., 2023c) | Multi-task metric eval | 847 samples | Factuality positive/negative | GitHub |

Table 2: Overview of AHE datasets/benchmarks for applications and meta-evaluation. "HF" indicates "HuggingFace".

**Technical, Scientific, and Industrial Domains** Beyond high-stakes professional services, AHE is being tailored for a range of specialized technical, scientific, and industrial applications. In software engineering, benchmarks like Collu-Bench (Jiang et al., 2024) are designed to detect hallucinations in code generation, a critical task in modern development workflows. In the realm of scientific discovery (AI4Science), datasets have been developed to identify "molecular mirages", hallucinations specific to LLM-based molecular comprehension in chemistry (Li et al., 2025a). Furthermore, evaluation is moving into specific industrial contexts, with benchmarks using spatio-temporal data to assess hallucinations in the analysis of traffic incidents, highlighting the unique challenges of applying LLMs to real-time, structured industry data (Li et al., 2025b).

### 2.4 Meta-Evaluation Benchmarks

Building on the large amount of automatic evaluation metrics, a number of specialized "meta-benchmarks" have emerged to systematically reassess and compare these metrics' abilities to detect hallucinations across different NLG tasks.

**Task-Specific Meta-benchmarks** Many early and prominent meta-benchmarks concentrate on a single, well-defined NLG task, most notably summarization and dialogue. For summarization, influential benchmarks like SummEval (Fabbri et al., 2021), GO FIGURE (Gabriel et al., 2021), SummaC (Laban et al., 2022), DialSumMeval (Gao & Wan, 2022), and AGGREFACT (Tang et al., 2023) assemble collections of human-annotated summaries to assess how well different metrics capture factual consistency. Furthermore, Adams et al. (2023) provide a meta-evaluation for long-form hospital-course summarization, which examines metric performance in a complex, domain-specific setting. In parallel, for dialogue systems, a suite of datasets such as FaithDial (Dziri et al., 2022a), Wizard of Wikipedia (Dinan et al., 2019), and others (Qin et al., 2021; Dziri et al., 2022b; Gopalakrishnan et al., 2019) provide turn-level annotations to directly evaluate whether model responses remain grounded in the provided knowledge.

**Broad-Coverage Meta-benchmarks** To assess the robustness and generalizability of AHE metrics, another category of benchmarks extends the evaluation to multiple domains and tasks. Some works bridge different scenarios. For instance, RealHall (Friel & Sanyal, 2023) connects closed- and open-domain settings to benchmark both SF and WF. BUMP (Ma et al., 2023) constructs minimal pairs-pairs of texts differing by a single, controlled factual error, to precisely test metric sensitivity. FELM (Chen et al., 2023c) further diversifies the landscape by incorporating complex reasoning tasks like scientific explanation and mathematical problem-solving. Another important line of work assesses metric robustness through adversarial testing. Subtle, hard-to-catch errors are created to probe the reliability of evaluators, such as for summarization (Chen et al., 2021) and RAG systems (Yu et al., 2024b). Generalist frameworks like TRUE (Honovich et al., 2022) and BEAMetrics (Scialom & Hill, 2021) evaluate metric performance across a wide spectrum of NLG tasks, from summarization to code generation. Critically, existing AHE metrics often show weak inter-correlation and lack consistency across different datasets (Kulkarni et al., 2025), leading to a important reliability challenge where a model's perceived performance can depend more on the chosen metric than on its actual capabilities.

### 2.5 Automated Dataset Generation

Manual annotation of hallucination benchmarks is labor-intensive, expensive, and difficult to scale. To overcome this bottleneck, a notable trend is the automated generation of benchmarks using LLMs themselves. This approach encompasses a spectrum of methodologies, evolving from simpler data augmentation techniques to highly sophisticated, controllable pipelines.

Early or simpler strategies focus on creating non-factual text through direct manipulation, such as random word seeding (Soleimani et al., 2023) or generating plausible substitutions by masking and refilling key information (Lee et al., 2022). However, the main focus of recent research is the pursuit of greater control and realism. This is achieved through structured pipelines that can inject specific, pre-defined error types, like distinguishing between intrinsic and extrinsic hallucinations (Utama et al., 2022). More broadly, a common technique is to use instruction-following LLMs to transform factual seed data into diverse, large-scale benchmarks with targeted hallucination types (Cao et al., 2023; Zhang et al., 2025b). At a more systemic

level, this automation can even involve multi-agent frameworks where LLM agents collaborate to turn unstructured text into verifiable evaluation items (Tong et al., 2025). The common goal driving this evolution is the creation of higher-quality, realistic errors that better mimic the subtle failures of modern LLMs.

While these automated methods enable the rapid creation of large-scale benchmarks, facilitating more extensive training and evaluation, they also introduce challenges. A key concern is ensuring the quality, subtlety, and real-world applicability of the generated errors, as purely model-generated data may contain biases or artifacts from the generator model itself (Yin et al., 2023).

### 2.6 Summary

We provide a comprehensive overview of the datasets in Table 1 and Table 2, including metadata and access links for reference and reproducibility. As a whole, the landscape of AHE benchmarks is rapidly maturing, moving from simple, task-specific annotations to granular, application-focused, and multilingual resources. However, challenges related to dataset scale, annotation consistency, and generalizability remain.

To address these gaps, future efforts in dataset construction should prioritize scalability, diversity of data sources, and the adoption of unified annotation schemas. This will be crucial for developing more reliable and universally applicable AHE methods. These datasets and benchmarks, despite their limitations, provide the essential foundation for developing and testing AHE methodologies. Having surveyed the landscape of what to evaluate on, we now turn our focus to how these evaluations are performed.

## 3 A Taxonomy of AHE Methodologies

We organize these methods into three core paradigms based on their fundamental operating principles. These paradigms are: (1) **Reference-based Evaluation**, which compares generated text against a static source; (2) **Reference-free Evaluation**, which detects hallucinations by analyzing the model's own outputs and internal states; and (3) **LLM-based Evaluation**, which leverages the advanced capabilities of LLMs as the primary evaluation tool. The overall distribution of tasks and their corresponding evaluation methods before and after the LLM era can be seen in Figure 3.

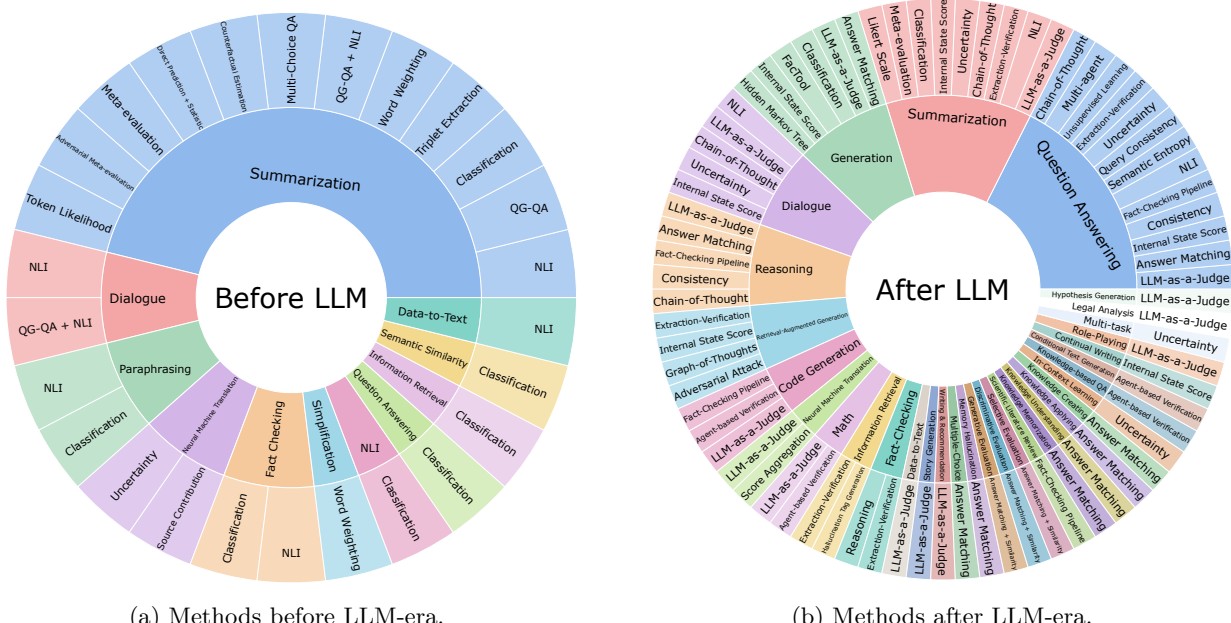

(a) Methods before LLM-era.                    (b) Methods after LLM-era.

Figure 3: Distribution of tasks and their corresponding methods before and after the LLM era.

### 3.1 Reference-based Evaluation

### 3.1.1 Stage 1: Evidence Collection Strategies

In this section, we focus on evidence collection strategies that do not rely on human-annotated ground truths. For SF evaluation, evidence is typically extracted directly from the input or surrounding context. In contrast, WF evaluation usually draws upon external resources or the model's own latent knowledge to verify the factual consistency of generated content.

**SF Evidence**   To evaluate SF, methods will extract salient information from the provided source document. A basic approach treats the **entire input as evidence**, which is viable for tasks with symmetric inputs like machine translation (Guerreiro et al., 2023; Dale et al., 2023a), but introduces significant noise in long-form tasks like summarization (Liu et al., 2022b). To avoid information redundancy in evidence collection, more recent methods employ strategies to **locate evidence in the input**, specifically targeting content that either supports or contradicts the output. A widely used approach in summarization evaluation is Question Generation and Question Answering (QG-QA), which generates questions from the output and verifies them against source-derived answers (Durmus et al., 2020; Wang et al., 2020). This paradigm has been refined through techniques like question weighting (Scialom et al., 2021) and exploring different QA model types (Fabbri et al., 2022). As an alternative to structuring evidence into QA pairs, another line of work focuses on decomposing the summary into a set of fine-grained, atomic claims, with each individual claim then being checked against the source document (Scirè et al., 2024; Zhang et al., 2024a). Beyond these paradigms, other approaches have emerged that represent source evidence in diverse forms, including semantic graphs (Ribeiro et al., 2022), discrete segments (Zha et al., 2023), or even hierarchical information structures that enable richer contextual modeling (Sun et al., 2025a). Further advancing this decompositional approach, Liu et al. (2025c) introduce a multi-stage pipeline to extract highly granular, context-aware atomic facts from both the source and the generated text, providing a more reliable foundation for the final verification step. This principle of finding fine-grained alignments also applied to structured data sources. For instance, when verifying text generated from tables, Perez-Beltrachini & Lapata (2018) use multi-instance learning to automatically discover correspondences between the data entries and segments of the text.

**WF Evidence**   To evaluate WF, evidence will be retrieved from external sources, which we categorize by their nature:

- **External Knowledge Base (KB):** This approach queries large, stable repositories of knowledge. Wikipedia is the most common source, from which facts are extracted in various formats, such as atomic facts (Min et al., 2023), entities (Yang et al., 2023), or triplets (Feng et al., 2023). The methodology extends to high-stakes domains by using specialized KBs like PubMed for medicine (Pal et al., 2023) or legal databases (Magesh et al., 2024b). A key challenge remains in selecting the most relevant evidence when multiple pieces are retrieved (Wang et al., 2024a).

- **LLM as KB:** An emerging trend is to use an LLM's own parametric knowledge, that is, the knowledge learned from training data and stored implicitly within the model's parameters, as a dynamic KB (Zheng et al., 2024a; Petroni et al., 2019). This involves prompting a powerful LLM to generate facts or knowledge that can serve as supplementary evidence for evaluation (Huang et al., 2024c; Chen et al., 2023a). While flexible, this approach carries an inherent risk of circular verification, where a model's own hallucination might be used to verify another lie.

- **Online Search:** To verify claims about recent or rapidly changing events, methods turn to real-time online search. The common workflow is to decompose a claim, issue targeted search queries, and synthesize the retrieved web snippets for a verdict (Chern et al., 2023). To enhance efficiency, refinements include pairing smaller models with search tools (Cheng et al., 2024) or adding a "check-worthiness" filter to avoid unnecessary queries for trivial or unverifiable claims (Wang et al., 2023c).

Finally, a comprehensive retrieval strategies aim to collect joint evidence for both SF and WF. These methods acknowledge that real-world outputs can be flawed in both ways simultaneously. Some approaches use a

unified framework to independently extract evidence from the source, external KBs, and the LLM's memory (Hu et al., 2024b). A more advanced methodology explicitly distinguishes between context-based (SF) and common-knowledge (WF) hallucinations within a single text, offering a more holistic and realistic evaluation framework (Paudel et al., 2025).

### 3.1.2 Stage 2: Comparison Techniques

Once evidence is prepared, the core evaluation happens at the comparison stage. These techniques range from lexical matching to semantic analysis.

**Lexicon-based Metrics**  These methods measure hallucination through word or phrase overlap. A common approach is to compute an Exact Match (EM) score for tokens, n-grams (Liu et al., 2022b) or fact triplets (entity-relation-entity) (Goodrich et al., 2019). Within Question Generation and Question Answering (QG-QA) frameworks, the F1-score is widely used to compare short, extracted answers (Durmus et al., 2020; Wang et al., 2020; Scialom et al., 2021), while other methods use statistical distances (e.g. KL-Div) over answers of multiple-choice questions (Manakul et al., 2023a). The lexical approach is also standard for scoring many QA-formatted benchmarks (Kasai et al., 2023; Oh et al., 2024; Lin et al., 2022), and has also been used to table-to-text generation through metrics like PARENT (Dhingra et al., 2019), which assess whether n-grams in the generated text are entailed by table content.

**Semantics-based Metrics**  To move beyond surface-level matching, semantic techniques are employed. The most prominent is Natural Language Inference (NLI), which reframes hallucination as an entailment problem: the evidence must logically entail the generated sentence or claim (Zhang et al., 2024b; Fabbri et al., 2022; Honovich et al., 2021). The effectiveness of NLI is often enhanced by representing text structurally, such as through dependency arcs (Goyal & Durrett, 2020) or semantic graphs (Ribeiro et al., 2022), and by fine-tuning NLI models on task-specific augmented data (Kryscinski et al., 2020; Feng et al., 2023). While these methods typically assume a natural language source, the NLI paradigm has also been adapted for structured data. In data-to-text generation, for example, the structured data source is first verbalized into a set of sentences, which can then serve as the premise for the NLI model to check against the generated text (Dušek & Kasner, 2020). Beyond binary entailment, more advanced methods perform multi-dimensional evaluation against richer error typologies (Xie et al., 2021; Muhlgay et al., 2024; Zha et al., 2023) or create a more robust judgment by aggregating multiple, diverse metrics (Wu et al., 2023; Himmi et al., 2024; Zhang et al., 2023b).

### 3.2 Reference-free Evaluation

In contrast to reference-based paradigms, reference-free evaluation dispenses with external grounding and instead assumes that hallucination signals can be detected through an "inward" examination of the model and its outputs. This line of approach is particularly valuable in scenarios where no reliable external reference is available.

**Consistency as a Proxy**  This line of inquiry operates on the premise that models tend to exhibit consistency, whereas hallucinations, as stochastic artifacts, are inherently unstable. The central idea is to elicit multiple outputs for a single input and use their degree of agreement as a proxy for reliability. Representative techniques include:

- **Comparing Multiple Sampled Responses:** This is the most direct approach, where multiple outputs are generated for the same prompt and their consistency is measured. For instance, SelfCheckGPT (Manakul et al., 2023b) measures consistency using textual similarity metrics, while KoLA (Yu et al., 2024a) uses a self-contrast score between two completions. A more advanced variant, EigenScore (Chen et al., 2024a), leverages the spectral properties (eigenvalues) of the responses' covariance matrix to quantify consistency.

- **Reconstruction-based Verification:** Instead of comparing outputs to each other, this technique assesses whether the original input query can be faithfully reconstructed from a generated response.

A high degree of similarity in reconstruction, as measured by InterrogateLLM (Yehuda et al., 2024), suggests that the model's output is a consistent and relevant elaboration of the input, rather than a hallucination.

**Uncertainty as a Proxy**   A parallel approach suggests that hallucinations arise when models exhibit reduced confidence. Such uncertainty, a salient signal for hallucination detection, can be estimated at different levels of the model architecture (Kadavath et al., 2022). Existing methods primarily concentrate on two dimensions:

- **Output Uncertainty:** The semantic information embedded in output representations can provide valuable signals for hallucination detection. This involves analyzing the final output layer of the model. It can be estimated directly from the log probabilities of generated tokens (Jesson et al., 2024; Son et al., 2022), or by examining the semantic properties of the output embeddings. The latter includes techniques like clustering or classifying response embeddings to identify outliers (Du et al., 2024; Su et al., 2024) or calculating the semantic entropy of the output distribution, which focuses on meaning rather than surface-form variation (Farquhar et al., 2024).

- **Internal Uncertainty:** This deeper analysis probes the model's latent representations for more subtle signals. Researchers have explored a wide array of techniques, such as training linear probes on hidden states to capture semantic entropy (Kossen et al., 2024), analyzing attention maps to detect contextual inconsistencies (Chuang et al., 2024; Sriramanan et al., 2024), and modeling the distributional distance of gradients (Hu et al., 2024c). In the context of RAG, this extends to analyzing the mechanisms related to both external and parametric knowledge by examining feed-forward layers and relevance propagation (Sun et al., 2025b; Hu et al., 2024a). Other methods focus on improving the interpretability of these signals through calibration or probabilistic modeling (e.g. decision tree, hidden Markov tree) (Zhou et al., 2025; Hou et al., 2024b).

Instead of relying on a single uncertainty signal, recent work explores combining multiple signals for more robust judgments. MetaCheckGPT (Mehta et al., 2024) presents this approach by integrating diverse uncertainty metrics (e.g., token probabilities, internal states) into a lightweight meta-model that learns their relationships and predicts hallucinations more reliably. In RAG-QA system, Gupta et al. (2025) quantify uncertainty by distinctly modeling signals from both the retrieval and the generation components, offering a more structurally-informed assessment of hallucination risk.

### 3.3   LLM-based Evaluation

In this section, we introduce approaches that leverage LLMs as evaluators for hallucination evaluation. The core premise of this approach is that LLMs possess parametric knowledge acquired during training and can be prompted to complete various tasks (Li et al., 2024a). Such methods can be further categorized into verbalized judge and judge with uncertainty, depending on whether the judgment is based on verbalized generation outputs or derived from internal model states, such as logits, attention maps, or layer-wise representations.

**LLM-as-a-Judge**   The evaluation process usually involves first providing the LLM with the evaluation criteria and task description, followed by supplying the task inputs for judgment. The feasibility of this approach, first systematically verified with tools like ChatGPT, demonstrated that LLMs can function as effective evaluators both with and without external references (Wang et al., 2023b). This flexibility allows for wide-ranging applications: it can be a reference-based judge in specific tasks, such as assessing faithfulness against a source document in long-form dialogue (Lattimer et al., 2023) or summarization (Jia et al., 2023; Wan et al., 2024), or a reference-free judge, relying on its own parametric knowledge (Chen et al., 2023b). This method has been scaled into general-purpose, multi-faceted frameworks that can integrate diverse knowledge forms (Fu et al., 2024; Liu et al., 2023; Zhang et al., 2024d) and produce fine-grained outputs, such as explicitly tagging hallucinated text spans instead of just providing a single score (Mishra et al., 2024a).

To move beyond a "black-box" judgment and enhance transparency, a crucial refinement has been the adoption of Chain-of-Thought (CoT) prompting (Liu et al., 2023; Friel & Sanyal, 2023; Akbar et al., 2024). This

reasoning can be used to provide detailed explanations and highlight inconsistent text for human review (Sreekar et al., 2024) or to enable complex, logic-based consistency checks on the reasoning process itself (Li et al., 2024b). Notably, COT can obscure common hallucination cues, as the step-by-step rationale often conveys spurious confidence, making errors more difficult to detect via uncertainty-based methods (Cheng et al., 2025).

**LLMs Cross-checking**  A more dynamic approach uses one LLM to cross-examine another, framing evaluation as an interactive verification process rather than a static scoring task. For example, in the $SAC^3$ framework (Zhang et al., 2023a), a verifier LM cross-checks the claims made by a generator LM, taking both input and output into account. In LMvLM (Cohen et al., 2023), an examiner LM poses follow-up questions to the generator, probing for inconsistencies in a conversational manner. This method is particularly adept at uncovering subtle or conditional hallucinations that static evaluation might miss.

**Integrated Frameworks**  The maturation of LLM-based methods is culminating in the development of extensive, open-source toolkits. Frameworks such as OpenFactCheck (Wang et al., 2024c), for example, provide a unified platform that integrates many of these LLM-based techniques, enabling researchers to conduct flexible and reproducible factuality evaluations.

### 3.4   Domain-Specific AHE Frameworks

While the LLM-based evaluators discussed previously offer powerful and flexible general-purpose capabilities, their effectiveness can be limited when confronted with the unique constraints, specialized knowledge, and high-stakes failure modes of specific domains (Pal et al., 2023). A generic evaluator, for example, may lack the domain-specific expertise to verify complex claims in a biomedical paper or to identify subtle logical errors in generated code. This has motivated research focused not on a single, universal evaluation method, but on developing specialized frameworks that adapt to meet these distinct challenges. In this section, we survey these emerging domain-specific AHE frameworks, which highlight the trend towards more contextualized and application-aware evaluation.

**High-Stakes Domains: Medicine and Law**  Beyond the specialized datasets for medicine and law described in subsection 2.3, a significant body of research has focused on developing AHE methods specifically tailored for these high-stakes domains. In medicine, research has focused on developing specialized metrics tailored to the nuances of medical language. For example, MedScore (Huang et al., 2025) is specifically designed to evaluate the factuality of free-form answers in medical QA, while PlainQAFact (You & Guo, 2025) is proposed to assess the hallucination of plain-language summaries of complex biomedical texts. Beyond proposing new metrics, researchers are also conducting deep evaluations of model capabilities on advanced tasks. Xiong et al. (2025), for instance, evaluated the truthfulness of LLMs in the complex task of biomedical hypothesis generation, probing the limits of scientific reasoning.

The legal domain presents similar challenges, demanding extreme precision and faithfulness to legal precedent and statutes. Research in this area has focused on both the practical assessment of existing tools and the fine-grained analysis of errors. Magesh et al. (2024a) assessed the reliability of leading commercial AI legal research tools, providing a real-world benchmark of their susceptibility to hallucination. Taking a more nuanced approach, Hou et al. (2024a) distinguished between "gaps" (information omission) and "hallucinations" (fabricated information) in legal analysis, arguing that not all inaccuracies are the same and that evaluation should be sensitive to error type.

**Behavioral and Systemic Consistency**  A second emerging frontier for AHE moves beyond real-world hallucination to evaluate consistency against defined behavioral or systemic rules. In role-playing applications, the primary failure mode is not factual inaccuracy but character hallucination, where a model breaks character or acts inconsistently with its defined persona. Research in this area ranges from evaluating a model's long-term memory and persona consistency over time (Ahn et al., 2024) to actively probing a model's resilience against interactive hallucinations, where a conflicting stance is deliberately introduced to manipulate the model's core beliefs (Kong et al., 2024).

This principle of behavior-based consistency extends to systemic domains like code generation. Here, hallucinations manifest as syntactically valid but semantically or logically incorrect code. Liu et al. (2024) explore and categorize these code hallucinations, highlighting the need for metrics that can verify logical correctness and faithfulness to API documentation. The concept evolves further in the context of complex, multi-agent, or multi-step reasoning. Evaluating these systems requires checking for logical failures over time, especially in dynamic scenarios with multi-round incomplete information (Dong et al., 2025). To ensure the reliability of the reasoning process itself, other frameworks propose the joint evaluation of both the final answer and the reasoning chain, verifying the internal consistency of the steps taken to reach a conclusion (Wang et al., 2025).

### 3.5 Summary

When ground truth or evidence is available, evaluation typically relies on measuring lexical or semantic similarity, where the NLI models can also integrate effectively with QG-QA evaluators. In the absence of references, intrinsic signals from the model serve as alternative proxies. LLMs also have emerged as particularly flexible evaluators, functioning as reference-based judges, reference-free arbiters leveraging parametric knowledge, or interactive cross-examiners probing subtle inconsistencies. Concurrently, domain-specific frameworks are increasingly developed to tailor evaluation to high-stakes and other practical fields. However, despite increasing confidence in LLMs as their size and capabilities expand, ensuring their stability and reliability in evaluation tasks remains an open challenge. Enhancing LLMs' capabilities in judgment, retrieval, and self-improvement represents a critical direction for future research.

## 4 Discussion

As shown in Figure 3, AHE methods have evolved from addressing early, specific tasks to tackling more complex, application-oriented challenges, a shift enabled by the increasing power of foundation models. Though many challenges have been addressed or mitigated by existing AHE methods, but there are still some questions that need to be investigated. Early methods addressed foundational challenges, but the increasing sophistication and deployment of LLMs in high-stakes, real-world applications necessitate a move beyond simple binary classifications of hallucination. This discussion critically analyzes the surveyed AHE methods through several key operational and theoretical lenses, highlighting both their capabilities and limitations. We organize our analysis around the core usability dimensions of AHE methods, the relationship between different hallucination types, and the gap between current evaluation paradigms and the complexities of real-world use cases.

### 4.1 Key Dimensions of AHE

An effective AHE method is not only accurate but also practical for its intended application. We find that existing approaches present a series of trade-offs across several usability dimensions, as detailed below.

**The Supervision Levels of AHE Methods** A key axis for categorizing AHE methods is their degree of supervision, which ranges across a spectrum from fully unsupervised to heavily supervised approaches. At one end, unsupervised, source-grounded methods like SelfCheckGPT (Manakul et al., 2023b) offer broad applicability. By operating without labeled data and assessing internal consistency, they are fast, cost-effective, and self-contained. However, their critical flaw is an inability to detect WF hallucinations where the model may be confidently and consistently wrong. In the middle of the spectrum lies weakly-supervised approaches, which aim to reduce the reliance on expensive human annotation. These methods often leverage heuristics, programmatic labeling, or other LLMs to generate noisy labels for fine-tuning (Wu et al., 2023). While more scalable than full supervision, their performance is capped by the quality of the heuristic labels. At the far end, fully-supervised methods (Chuang et al., 2024) are trained on human-annotated, domain-specific datasets. Although they can achieve high performance, they face two main challenges: the high cost of annotation and, more importantly, poor generalizability. A model fine-tuned to detect hallucinations in news summaries may fail spectacularly when applied to a different domain, as the linguistic patterns and types of errors would not transfer well (Thorne et al., 2018). This creates a challenging landscape where the

choice of supervision method should be carefully weighed against available resources, data, and the specific application's tolerance for different types of error.

**Granularity of Evaluation: The Myth of an Optimal Level**    The studies reviewed in this work evaluate hallucinations across various granularities, ranging from fine-grained units, such as individual tokens and entities, to more coarse-grained elements, including phrase spans, claims, sentences, and even document-level segments. This raises a critical question posed by the reviewer: is there an optimal level of granularity? We argue that the assumption of a single, universally optimal granularity is a misconception. The ideal level is intrinsically tied to the application's tolerance for error and its specific goals. For instance, in a medical summarization task, a single incorrect entity (e.g., "50mg" instead of "15mg") could have severe consequences, demanding fine-grained, entity-level evaluation. Conversely, in a creative writing context, sentence-level coherence and narrative flow may be far more important than the factual accuracy of individual claims. Early methods favored structured representations like entity-relation triplets, while recent approaches have shifted towards decomposing text into atomic facts or claims (Xie et al., 2021; Chen et al., 2023c). This finer-grained approach offers more precise error localization but can fail to capture sentence-level semantic errors or misleading implications. Thus, rather than seeking a single "best" granularity, future research should focus on multi-level, task-adaptive evaluation frameworks that can dynamically adjust their focus based on the application's specific needs.

**Aligning with Human Judgement**    While human evaluation serves as the ultimate gold standard for AHE, ensuring its consistency and reliability presents a non-trivial challenge, especially for the complex and often deceptive hallucinations produced by modern LLMs. To address this, it is crucial to develop clear evaluation criteria and unified annotation guidelines (van der Lee et al., 2019; Howcroft et al., 2020). Indeed, research has shown that methodological choices, such as adopting a finer granularity of judgment can reduce inter-annotator variance and improve the reliability of human faithfulness scores (Krishna et al., 2023). Meanwhile, to mitigate the high cost and workload of manual annotation, an alternative is to use powerful LLMs like GPT-4 as a proxy for human judgment (LLM-as-a-judge), especially in smaller-scale or lightweight AHE settings (Chuang et al., 2024). This approach offers greater scalability and convenience than full human evaluation but introduces its own challenges, as they may exhibit biases and influenced by prompting strategies. A crucial implication of this LLM-as-a-judge approach is that the LLM used for annotation effectively becomes the performance ceiling for the AHE methods being evaluated. A downstream evaluator can only aspire to replicate the judgments of its LLM teacher, but cannot surpass them in quality or uncover the teacher's inherent biases on that benchmark. Thus, while LLM-as-a-judge provides a practical tool for large-scale assessment, it ultimately represents a trade-off between scalability and the nuanced, gold-standard insights afforded by well-structured human evaluation.

## 4.2   The Evolving Landscape of Hallucination and Evaluation

The rapid progress of LLMs creates a persistent tension: as their capabilities grow, failure modes become increasingly subtle and complex, continually outpacing existing benchmarks and demanding constant reevaluation of reliability.

**The Changing Nature of Hallucinations**    A crucial question is how hallucinations in current models have evolved compared to those in older ones. Early models, such as encoder-decoder architectures (Devlin et al., 2019), often produced hallucinations that were more syntactically flawed, ungrammatical phrases or direct contradictions that were relatively simple to spot See et al. (2017). In contrast, modern foundation models are fluent and plausible, but their hallucinations are far more deceptive. Today's errors are often context-dependent and embedded in complex reasoning, such as making a logically sound argument from a single false premise or presenting a literally true statement in a misleading way. Moreover, hallucinations may emerge not from factual misprocessing, but from misalignment with user intent (Hao et al., 2025). This evolution means that detecting hallucinations is no longer just a task of fact-checking, but one of deep reasoning and pragmatic understanding of user intention.

**From Static Benchmarks to Active Probing**  Given the increasing subtlety of these new hallucinations, the field is moving beyond passive evaluation on static benchmarks towards more active probing to stress-test model reliability. This involves deliberately "attacking" an LLM to see when and how it hallucinates. One approach is to present the model with unanswerable questions, to see if it correctly expresses uncertainty or confidently fabricates an answer (Sun et al., 2024; Ouyang, 2025). Another powerful technique involves providing a false premise within the prompt and observing whether the model blindly accepts it and generates a logically consistent but entirely hallucinated scenario, or if it "fights back" by correcting the user's premise (Yuan et al., 2024). More broadly, adversarial probing, which systematically perturbs inputs to mislead the model, has also been employed to examine the robustness of both generation systems and evaluators (Yu et al., 2024b). These methods provide a deeper insight into a model's true reliability, assessing not just its factuality, but its intellectual honesty.

### 4.3 The Complexities of Hallucination in Real-World Applications

Perhaps the most significant weakness of the current AHE landscape is its failure to adequately model the nuances of hallucinations in real-world, high-stakes domains. A simple true/false dichotomy is dangerously insufficient. Future work should embrace the following complexities:

**Challenges in Specific Domains**  The practical deployment of AHE systems is constrained by real-world challenges, particularly concerning latency, cost, and the profound complexities of specialized domains. Methods that depend on external evidence, whether through search engine APIs or knowledge base queries, introduce latency and cost (Chern et al., 2023) that can be prohibitive for real-time applications like conversational agents. While leveraging powerful LLMs such as GPT-5 for evaluation can be effective, it also has computational constraints, rendering such methods more suitable for offline batch processing or post-hoc content auditing than for interactive use (Liu et al., 2023). Yet, the most pressing challenge lies in domain specificity, as the consequences of hallucination are highly context-dependent.

- **Medical:** A hallucination in the medical application is not merely a factual error but a potential risk to patient safety. Medical data are inherently heterogeneous, including charts, imaging, and clinical documentation, while also relying on specialized terminology and being subject to privacy constraints, making high-quality datasets scarce. Evaluation in this context requires a graded assessment that accounts for both the severity and clinical implications of errors, differentiating between trivial inaccuracies and potentially life-threatening mistakes (Singhal et al., 2023).

- **Legal:** Faithfulness requires precise interpretation of statutes and precedents in legal domain. A hallucination could involve citing a non-existent case law or misrepresenting a legal principle, with serious legal consequences (Cui et al., 2023). The "evidence" itself is dense, argumentative text, making verification a task of deep reasoning rather than simple fact retrieval.

The medical and legal fields are merely two high-stakes applications that have received attention. In many other fact-critical scientific domains, such as materials science (Lei et al., 2024), drug discovery (Zheng et al., 2024b), and climate science (Diggelmann et al., 2020), developing robust hallucination detection methods still demands further investigation. These examples illustrate that a one-size-fits-all approach to AHE is untenable so far. The pursuit of higher factual accuracy should be balanced not only against latency and cost but also against the deep, domain-specific requirements for data, reasoning, and evaluation criteria.

**The Ambiguous Boundary: Hallucination, Abstraction, and Imagination**  A critical weakness in the current AHE paradigm is its tendency to treat all deviations from source or fact as errors. This overlooks the nuanced reality that in many real-world applications, such deviations are not only acceptable but intentional and desirable. For instance, sometimes factual hallucination is good (Cao et al., 2022). In legal summarization, a model that incorporates accurate, external legal principles to provide context is beneficial, which performs knowledge-informed abstraction. This act deliberately sacrifices strict SF to enhance the summary's utility and WF, representing a feature of advanced reasoning, not a bug. Similarly, in creative domains, labeling imaginative content, such as drafting a poem or brainstorming a fictional story, as a

"hallucination" is a fundamental category error. In these contexts, the model is fulfilling its intended purpose, where the metric of success is creativity or novelty, not factual accuracy. Therefore, a mature AHE framework should be context-aware, capable of distinguishing between an unintentional factual error and a controlled, task-appropriate deviation from verifiable facts. The ultimate goal is not to simply penalize any divergence, but to evaluate whether the generated output aligns with the user's intent and the specific requirements of the domain (Zhou et al., 2024).

**Reframing the Goal: Towards Controllable SF and WF**   The crucial distinction between unintentional error and intentional deviation logically reframes the ultimate research goal: it should evolve from merely detecting hallucinations to enabling controllable generation across the axes of SF and WF. This new paradigm treats SF and WF not as static, universal virtues, but as dynamic, task-dependent variables that a model learns to balance. The ideal generative model is not one that is simply "factual," but one that can operate at various points along this two-dimensional spectrum as required, be it high-SF/low-WF for a perfect summary of a story source, or high-SF/high-WF for the knowledge-informed abstraction needed in legal analysis. Recent advances in instruction tuning and model editing represent emerging but promising steps in this direction. Techniques that fine-tune models on nuanced commands provide a mechanism for explicitly requesting either strict faithfulness or creative synthesis (Zhou et al., 2023). Meanwhile, the field of model editing offers a more surgical approach to correcting WF errors or implanting new knowledge directly into a model's parameters, representing a form of post-hoc factual control (Wang et al., 2024b). This evolution in model capabilities presents a new mandate for AHE. The next generation of evaluation benchmarks should therefore assess not just binary accuracy (Jing et al., 2025), but a model's capacity to modulate its output along the SF/WF spectrum in alignment with explicit instructions.

## 5   Future Directions

While existing AHE methods have demonstrated substantial progress, critical gaps persist in hallucination detection and evaluation. Particularly in cutting-edge task domains, certain hallucinations remain complex and difficult to detect and evaluate, which deserve further investigation.

**Interpretability**   Previous hallucination evaluation efforts primarily focused on model outputs rather than underlying mechanisms. However, analyzing factual granularity and underlying causes can substantially enhance our understanding of these phenomena. Future research directions show significant promise across multiple fronts. Reasoning-based approaches (Liu et al., 2025a; Akbar et al., 2024) demonstrate potential for uncovering hallucination origins and providing more informative evaluations. Emerging studies investigate leveraging internal model states for assessment (Chuang et al., 2024; Hu et al., 2024c; Su et al., 2024), examining how the origin, distribution, and layer-wise dynamics of neural representations relate to hallucination phenomena. Advanced interpretability methods, including sparse autoencoder(SAE)-based approaches, attempt to project neurons into more interpretable spaces for systematic analysis. These internal mechanism investigations represent a critical frontier, as the fundamental drivers of hallucinations remain poorly understood and offer substantial opportunities for breakthrough insights into model reliability.

**Complex Context**   Effectively addressing hallucinations arising from a model's difficulty in processing complex inputs, such as long or multi-format contexts, is of critical importance. Current research on LLMs in long-context scenarios primarily focuses on handling extended input sequences, for instance, incorporating entire contexts or multi-turn dialogue histories. However, hallucinations from inconsistencies within long outputs, particularly contradictions between the beginning and the end of a generated text, remain underexplored (Wei et al., 2024b), such as detecting inconsistencies in character behavior within model-generated narratives. Semantically, complexity arises from pragmatic nuance, where detecting literally true but contextually misleading statements remains a challenge, requiring deeper world knowledge and inferential reasoning than current benchmarks assess. To enhance robustness in such scenarios, the incorporation of multi-evidence verification into hallucination evaluation offers a promising research direction (Wang et al., 2024a), as it can enhance the robustness of factuality assessments by grounding model outputs against multiple corroborating sources.

**Efficiency**   Looking ahead, improving the efficiency of hallucination evaluation will be critical for enabling large-scale, real-time assessment of generated text. First, the challenge for efficient evaluation is moving beyond simply distilling AHE scores (Rajendhran et al., 2025; Belyi et al., 2025). The more forward-looking goal is to develop lightweight models that can perform deeper assessments, such as predicting hallucination severity or verifying domain-specific rules. Second, the integration of multi-granular caching and incremental evaluation pipelines would allow reasonable allocation of compute resources. Third, by building upon principles from human-in-the-loop settings, future AHE systems can be made far more efficient. For instance, systems could be developed to automatically estimate the potential severity of a detected hallucination, ensuring that expert human review is reserved only for the highest-risk cases, thereby optimizing the cost and impact of manual oversight (Schiller, 2024). Together, these directions would move hallucination evaluation from an expensive research, only procedure toward a practical component of everyday NLG pipelines.

**Hallucinations in Emerging Domains**   Recent research has expanded LLMs into diverse domains, including multilingual communication, multimodal understanding, and autonomous systems, introducing novel hallucination types distinct from traditional text generation. These include code hallucination, syntactically valid but semantically incorrect code (Qian et al., 2023); tool hallucination from false assumptions about external tool behavior (Zhang et al., 2024e); visual hallucination involving inaccurate content descriptions (Huang et al., 2024a); cross-lingual hallucination where meaning distorts across languages (ul Islam et al., 2025; Kang et al., 2024); and multimodal hallucination featuring cross-modal inconsistencies (Huang et al., 2024b). These domain-specific challenges require specialized evaluation frameworks that can transfer knowledge across contexts while addressing unique characteristics of each application domain. Developing robust evaluation methods for these emerging hallucination forms proves both intellectually compelling and essential for ensuring LLM system reliability and safety in diverse application contexts.

## 6   Conclusion

Hallucinations in Natural Language Generation (NLG) represent a fundamental threat to the reliability, safety, and applicability of language models. Consequently, developing robust evaluation methods is not merely a diagnostic exercise, but for mitigating risks, advancing model development, and fostering user trust.

In this survey, we have systematically reviewed recent advances in the field of automatic hallucination evaluation, structuring our discussion about datasets and methodologies. This includes both the source faithfulness and world factuality, which differ in their grounding requirements and thus present distinct evaluation challenges.

Historically, the majority of hallucination evaluation methods have been designed in a task-specific manner, as defining clear performance criteria is often more straightforward within narrowly scoped applications. However, the rise of LLMs has brought new demands and exposed limitations in existing approaches. These models are typically deployed in open-ended, multi-domain contexts, where traditional task-based metrics fall short in capturing nuanced hallucinations. Consequently, the community has been driven to reconsider and refine existing evaluation paradigms, aiming to develop more generalizable and scalable evaluation frameworks.

Going forward, addressing hallucination in LLMs will require continued efforts in both benchmark construction and metric development, particularly those that are sensitive to domain-specific knowledge, real-world reasoning, and the dynamic nature of factual correctness. Future breakthroughs would likely emerge from integrating insights from fields including knowledge representation, model interpretability, and robust human-in-the-loop evaluation. As such, hallucination evaluation is not merely a downstream concern but a foundational issue that will define the next generation of trustworthy NLG systems.

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

## A  Paper Inclusion Criteria

The literature screening process for this study adhered to the PRISMA guidelines (Haddaway et al., 2022). The flow diagram below illustrates the identification, screening, and inclusion phases. Our initial search identified 223 records from DBLP databases and 147 records from other methods such as websites (ACL Anthology and Google Scholar), organisations (ICLR, ICML, NeurIPS, and AAAI), and citation searching. From the database search, 58 duplicate records were removed before screening. In the screening phase, the remaining 165 records were reviewed, and 27 were excluded because of the duplication from other sources. Subsequently, 138 papers were sought for retrieval and being assessed for eligibility. From this group, 66 papers were excluded for focusing on hallucination mitigation (n=5) or multi-modality (n=61). The 147 papers from other sources were also assessed for eligibility while searching, with none being excluded at this stage. Ultimately, this comprehensive process resulted in 219 studies being included in the final review.

## B  Comparing SF and WF Evaluation

SF and WF evaluations are two subcategories of AHE. While they assess hallucination from different perspectives, they also share certain points of convergence. For example, some evaluation methods in both SF and WF rely on reference texts for comparison in order to produce a final judgment. Moreover, both types of evaluation can be conducted by analyzing the model's internal states. In this section, we present a case study on how the evaluation of SF and WF may influence each other when using the same evaluator, by presenting the evaluation results of four SF and WF evaluators across the four quadrants shown in Figure 1.

The cases presented here are based on the summarization data shown in Table 3, which are drawn from the XEnt dataset (Cao et al., 2022) and FactCollect (Ribeiro et al., 2022). We selected four evaluators representing different perspectives, including both GPT-based (SelfCheckGPT, HaluEval, FacTool) and non-GPT-based models (WeCheck), and covering evaluators designed for assessing both SF and WF aspects. SelfCheckGPT uses a zero-shot approach in its prompt to assess the consistency, HaluEval's prompt provides examples for judgment, and FacTool aggregates online search to judge the factuality. For GPT-based models, we specifically used GPT-3.5-turbo. Although FacTool is not originally designed for summarization evaluation, we adapted it to the KBQA (Knowledge-Based Question Answering) setting in order to explore its transferability to this task. All the evaluators only provide binary classification results.

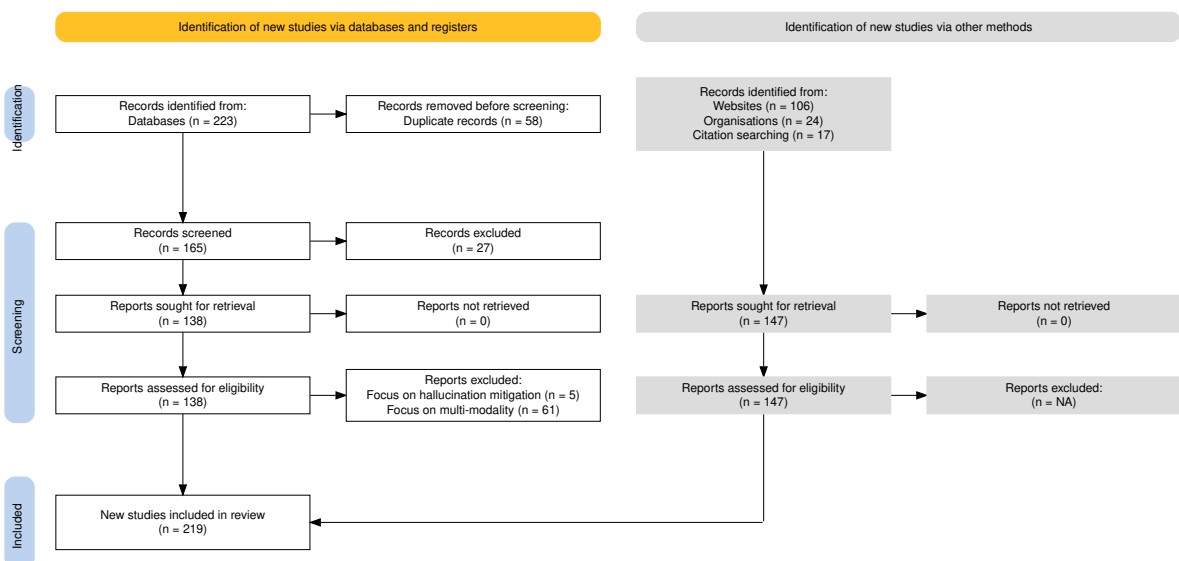

Figure 4: PRISMA flow diagram.

The results of different models on these cases show considerable variation. The SFE cases indicate that the results of SelfCheckGPT and HaluEval remain unstable. For the WFE cases, FacTool provides the correct answers, and surprisingly, WeCheck also made correct judgments. This result aligns with Qi et al. (2025), which suggests that the model's ability in one aspect may subconsciously influence its evaluation in the other. In other words, SF and WF evaluations can affect each other, primarily due to the presence of misaligned information within the model.

## C   Supplementary Tables

### C.1   Evaluator Meta Information

We present a set of tables summarizing the meta-information of the surveyed evaluators, as shown in Table 4, Table 5, and Table 6. In the *New Dataset* column, if the dataset name is identical to the evaluator's name, it indicates that the authors did not explicitly name the dataset; instead, we assign the evaluator's name for clarity and reference. The *Based-model* column refers to the underlying models used by each evaluator either for performing evaluation or for generating synthetic data. The *Method* column describes the evaluation pipeline, methodological framework, or the primary novel contribution introduced by the evaluator. The *Metric* column specifies the scoring strategy or computational approach used to produce the final evaluation score. Lastly, the *SF* (Source Faithfulness) and *WF* (World Factuality) columns use ✓ and ✗ to indicate whether an evaluator explicitly addresses each respective aspect.

| | Document | Summary | Note | WeCheck | SelfCheckGPT | HaluEval | FacTool |
|---|---|---|---|---|---|---|---|
| SF-WF | ... Harry Kane has been given the nod by Youssouf Mulumbu for this season's players' Player of the Year award The West Brom midfielder has picked Chelsea wideman Eden Hazard for the young player of the year prize Congo international Mulumbu posted his votes for this year's PFA awards to Twitter on Wednesday Mulumbu challenges QPR defender Yun Suk-Young during West Brom's 4-1 defeat at The Hawthorns Goalkeepe ... | The DR Congo international has picked Chelsea wideman Eden Hazard for the young player of the year prize . | The summary is correct. | TRUE | TRUE | TRUE | FALSE |
| SF-WFE | ... Since the end of March, the Vikings' only wins have been in the Challenge Cup against lower-league sides. "We've got the personnel and we've got the people to spark us back into life," **Chris Betts** told BBC Radio Merseyside. "When we get rolling again I'm sure, or I'm positive, that we can really turn this year around for ourselves." ... "The players are hurting and we've got to win," added England assistant coach Betts. ... | Widnes Vikings can turn their poor start to the Super League season around if they can find a winning streak, says assistant coach **Chris Betts**. | "Chris Betts" is in the document but is incorrect essentially. | FALSE | TRUE | TRUE | FALSE |
| SFE-WF | The panther chameleon was found on Monday by a dog walker in the wooded area at **Marl Park**. It had to be put down after X-rays showed all of its legs were broken and it had a deformed spine. RSPCA Cymru said it was an "extremely sad example of an abandoned and neglected exotic pet". ...... | A chameleon has been put down by RSPCA Cymru after it was found injured and abandoned in a **Cardiff park**. | The Marl Park is in Cardiff but not mentioned in the document. | TRUE | FALSE | TRUE | TRUE |
| SFE-WFE | A number of men, two of them believed to have been carrying guns, forced their way into the property at Oakfield Drive shortly after 20:00 GMT on Saturday. They demanded money before assaulting a man aged in his 50s. ... Alliance East Antrim MLA Stewart Dickson has condemned the attack. ... | A man has been assaulted by a gang of armed men during a robbery at a house in **Ballymena**, County Antrim. | "**Ballymena**" is neither in the document nor correct according to external knowledge. | FALSE | TRUE | TRUE | FALSE |

Table 3: Examples of the results from selected evaluators on the ==Source Faithfulness Error (SFE) and World Factuality Error (WFE)==. "TRUE" means the evaluator labeled it as correct while "FALSE" means incorrect.

| Era | Name | New Dataset | Data Source | Fact Definition | Task | Based-model | Method | Metric | SF | WF |
|---|---|---|---|---|---|---|---|---|---|---|
| Before LLM Era | $Fact_{acc}$ | WikiFact | Wikipedia, Wikidata KB | Triplet | Summ | Transformer | Triplet Extraction | P, R, F1 | ✓ | ✗ |
| | FactCC | FactCC | CNN/DM, XSumFaith | Sent | Summ | BERT | NLI (2-class) | Likelihood | ✓ | ✗ |
| | DAE | DAE | PARANMT50M | Dependency | Summ | ELECTRA | NLI (2-class) | Likelihood | ✓ | ✗ |
| | Maskeval | / | CNN/DM, WikiLarge, ASSET | Word | Summ, Simp | T5 | Word Weighting | Weighted Match Score | ✓ | ✗ |
| | Guerreiro et al. (2023) | Haystack | WMT2018, DE-EN | Text Span | NMT | Transformer | Uncertainty Measure | Avg. Similarity | ✓ | ✗ |
| | Dale et al. (2023a) | / | Haystack | Text Span | NMT | Transformer | Source Contribution | Percentage | ✓ | ✗ |
| | FEQA | FEQA | CNN/DM, XSum | Sent Span | Summ | BART (QG), BERT (QA) | QG-QA | Avg. F1 | ✓ | ✗ |
| | QAGS | QAGS | CNN/DM, XSum | Ent, Noun Phrase | Summ | BART (QG), BERT (QA) | QG-QA | Avg. Similarity | ✓ | ✗ |
| | QuestEval | / | CNN/DM, Xsum | Ent, Noun | Summ | T5 (QG, QA) | QG-QA | P, R, F1 | ✓ | ✗ |
| | QAFactEval | / | SummaC | NP Chunk | Summ | BART (QG), ELECTRA (QA) | QG-QA, NLI | LERC | ✓ | ✗ |
| | MQAG | / | QAGS, XSumFaith, Podcast, Assessment, SummEval | Sent Span | Summ | T5 (QG), Longformer (QA) | Multi-Choice QA | Choice Statistical Distance | ✓ | ✗ |
| | CoCo | / | QAGS, SummEval | Token, Span, Sent, Doc | Summ | BART | Counterfactual Estimation | Avg. Likelihood Diff | ✓ | ✗ |
| | FactGraph | FactCollect | CNN/DM, XSum | Dependency | Summ | ELECTRA | Classification | BACC, F1 | ✓ | ✗ |
| | FactKB | FactKB | CNN/DM, XSum | Triplet | Summ | RoBERTa | Classification | BACC, F1 | ✓ | ✗ |
| | ExtEval | ExtEval | CNN/DM | Discourse, Coreference, Sentiment | Summ | SpanBERT, RoBERTa | Direct Prediction, Statistic | Summation of Sub-scores | ✓ | ✗ |
| | $Q^2$ | $Q^2$ | WOW | Sent Span | Diag | T5 (QG), Albert-Xlarge (QA), RoBERTa (NLI) | QG-QA, NLI | Likelihood | ✗ | ✓ |
| | FactPush | / | TRUE | Text Span | Diag, Summ, Paraphrase | DeBERTa | NLI | AUC | ✓ | ✗ |
| | AlignScore | / | 22 datasets from 7 tasks | Sent | NLI, QA, Paraphrase, Fact Verification, IR, Semantic Similarity, Summ | RoBERTa | 3-way Classification | Likelihood | ✓ | ✗ |
| | WeCheck | / | TRUE | Response | Summ, Diag, Para, Fact Check | DeBERTaV3 | Weakly Supervised NLI | Likelihood | ✓ | ✗ |
| | PARENT | / | WIKIBIO | Attribute | Table2Text | LSTM-based | Parent-based Scoring | PARENT (P,R,F1) | ✓ | ✗ |
| | Perez-Beltrachini & Lapata (2018) | / | DBPedia, Wikipedia | Triplet | Data2Text | Encoder-decoder | Multi-instance Learning | BLEU, ROUGE | ✓ | ✗ |
| | Dušek & Kasner (2020) | / | Data-to-text | Attribute | Data2Text | BERT | Lexicon-NLI | Accuracy | ✓ | ✗ |
| | GO FIGURE | / | CNN/DM, XSum | Response | Summ | BERT, RoBERTa | Meta-evaluation | Pearson, Spearman | ✓ | ✗ |
| | Chen et al. (2021) | / | CNN/DM, XSum | Response | Summ | Multiple | Adversarial Meta-eval | ASR | ✓ | ✗ |
| | HaRiM+ | / | CNN/DM, XSum | Word | Summ | PLMs | Token Likelihood | Pearson, Spearman | ✓ | ✗ |

Table 4: AHE Meta-Info Table before LLM era, which means the methods do not rely on the ability of LLMs such as ChatGPT.

| Era | Name | New Dataset | Data Source | Fact Definition | Task | Based-model | Method | Metric | SF | WF |
|---|---|---|---|---|---|---|---|---|---|---|
| After LLM Era | SCALE | ScreenEval | LLM, Human | Sentence | Long Diag | Flan-T5 | NLI | Likelihood | ✓ | ✗ |
| | Chen et al. (2023b) | / | SummEval, XSumFaith, Goyal21, CLIFF | Response | Summ | Flan-T5, code-davinci-002, text-davinci-003, ChatGPT, GPT-4 | Vanilla/COT/ Sent-by-Sent Prompt | Balanced Acc | ✓ | ✗ |
| | GPTScore | / | 37 datasets from 4 tasks | Various | Summ, Diag, NMT, D2T | GPT-2, OPT, FLAN, GPT-3 | Direct Assessment | Direct Score | ✓ | ✗ |
| | G-Eval | / | SummEval, Topical-Chat, QAGS | Response | Summ, Diag | GPT-4 | COT, Form-filling | Weighted Scores | ✓ | ✗ |
| | Wang et al. (2023b) | / | 5 datasets from 3 tasks | Response | Summ, D2T, Story Gen | ChatGPT | Direct Assessment, Rating | Direct score | ✓ | ✗ |
| | ChainPoll | RealHall-closed, RealHall-open | COVID-QA, DROP, Open Ass prompts, TriviaQA | Response | Hallu Detect | gpt-3.5-turbo | Direct Assessment (2-class) | Acc | ✓ | ✗ |
| | EigenScore | / | CoQA, SQuAD, TriviaQA Natural Questions | Inner State | Open-book QA Closed-book QA | LLaMA, OPT | Semantic Consistency/ Diversity in Dense Embedding Space | AUROC, PCC | ✓ | ✗ |
| | TruthfulQA | TruthfulQA | LLM, Human | Response | Multi-Choice QA, Generation | GPT-3-175B | Answer Match | Percentage, Likelihood | ✗ | ✓ |
| | HaluEval | Task-specific, General | Alpaca, Task datasets ChatGPT | Response | QA, Summ, Knowledge-grounded Diag, Generation | ChatGPT | Direct Assessment | Acc | ✓ | ✓ |
| | FACTOR | Wiki-/News-/ Expert- FACTOR | Wikipedia, Refin-edWeb, ExpertQA | Sent Span | Generation | / | FRANK Error Classification | likelihood | ✗ | ✓ |
| | FELM | FELM | TruthfulQA, Quora, MMLU, GSM8K, ChatGPT, Human | Text Span, Claim | World Knowledge, Sci and Tech, Math, Writing and Recom-mendation, Reasoning | Vicuna, ChatGPT, GPT4 | Direct Assessment | F1, Balanced Acc | ✓ | ✓ |
| | FreshQA | Never/Slow Fast-changing, false-premise | Human | Response | Generation | / | Answer Match | Acc | ✗ | ✓ |
| | RealTimeQA | RealTimeQA | CNN, THE WEEK, USA Today | Response | Multi-Choice QA, Generation | GPT-3, T5 | Answer Match | Acc, EM, F1 | ✗ | ✓ |
| | ERBench | ERBench Database | 5 datasets from Kaggle | Ent-Rel | Binary/ Multiple -choice QA | / | Direct Assessment, String Matching | Ans/Rat/ Ans-Rat Acc, Hallu Rate | ✗ | ✓ |
| | FactScore | / | Biographies in Wikipedia | Atomic Fact | Generation | InstructGPT, ChatGPT, PerplexityAI | Binary Classification | P | ✗ | ✓ |
| | BAMBOO | SenHallu, AbsHallu | 10 datasets from 5 tasks | Response | Multi-choice tasks, Select tasks | ChatGPT | Answer Match | P, R, F1 | ✓ | ✗ |
| | MedHalt | MedHalt | MedMCQA, Medqa USMILE, Medqa (Taiwan), Headqa, PubMed | Response | Reasoning Hallu Test, Memory Hallu Test | ChatGPT | Answer Match | Pointwise Score, Acc | ✗ | ✓ |
| | ChineseFactEval | ChineseFactEval | / | Response | Generation | / | FacTool, Human annotator | Direct Score | ✗ | ✓ |
| | UHGEval | UHGEval | Chinese News Websites | Keywords | Generative/ Discriminative/ Selective Evaluator | GPT-4 | Answer Match, Similarity | Acc, Similarity Score | ✗ | ✓ |
| | HalluQA | HalluQA | Human | Response | Generation | GLM-130B, ChatGPT, GPT-4 | Direct Assessment | Non-hallu Rate | ✗ | ✓ |
| | FacTool | / | RoSE, FactPrompts, HumanEval, GSM-Hard, Self-instruct | Claim, Response | Knowledge-based QA, Code Generation, Math Reasoning, Sci-literature Review | ChatGPT | Claim Extraction, Query Generation, Tool Querying, Evidence Collection, Agreement Verification | P, R, F1 | ✓ | ✓ |
| | UFO | / | NQ, HotpotQA, TruthfulQA, CNN/DM, Multi-News, MS MARCO | Ent | Open-domain/ Web Retrieval-based/ Expert-validated/ Retrieval-Augmented QA, News Fact Generation | gpt-3.5-turbo-1106 | Fact Unit Extraction, Fact Source Verification, Fact Consistency Discrimination | Avg. Sub-scores | ✓ | ✓ |
| | CONNER | / | NQ, WoW | Sentence | Open-domain QA, Knowledge-grounded Dialogue | NLI-RoBERTa -large, ColBERTv2 | 3-way NLI | Acc | ✗ | ✓ |
| | SelfCheckGPT | SelfCheckGPT | WikiBio | Response | Hallu Detect | GPT-3 | NLI, Ngram, QA, BERTScore, Prompt | AUC-PR | ✓ | ✗ |
| | InterrogateLLM | / | The Movies Dataset, GCI The Book Dataset (Kaggle) | Response | Hallu Detect | GPT-3, LLaMA-2 | Query Consistency | AUC, Balanced Acc | ✗ | ✓ |
| | $SAC^3$ | / | HotpotQA, NQ-open | Response | QA Generation | gpt-3.5-turbo, Falcon-7b-instruct, Guanaco-33b | Cross-checking, QA Pair Consistency | AUROC | ✓ | ✓ |
| | KoLA | KoLA | Wikipedia, Updated News and Novels | Response | Knowledge Memorization /Understanding/Applying /Creating | / | Self-contrast Answer Match | Similarity | ✗ | ✓ |
| | RV | PHD | Human Annotator | Ent | Generation | ChatGPT | Construct Query, Access Databases, Entity-Answer Match | P, R, F1 | ✓ | ✗ |
| | SummEdits | SummEdits | 9 datasets from Summ task | Span | Summ, Reasoning | gpt-3.5-turbo | Seed summary verify, Summary edits, Annotation | Balanced Acc | ✓ | ✗ |
| | LLM-Check | / | FAVA-Annotation, RAGTruth, SelfcheckGPT | Response | Fact-checking | Llama-2, Llama-3, GPT4. Mistral-7b | Analyze internal attention kernel maps, hidden activations and output prediction probabilities | AUROC, FPR, Acc | ✗ | ✓ |
| | PHR | synthetic | / | Response | ICL | Llama-2, Gemma-2 | Posterior Hallucination Rate (Baysian) | Hallu Rate | ✓ | ✗ |
| | HalluMeasure | TechNewsSumm | CNN/DM, SummEval | claim | Summ | Claude | COT, Reasoning | P, R, F1 | ✓ | ✗ |
| | EGH | / | HADES, HaluEval, SelfcheckGPT | Response | QA, Diag Summ | LLaMa2, OPT, GPT-based | Taylor expansion on embedding difference | Acc, P, R, F1, AUC, G-Mean, BSS | ✓ | ✓ |
| | STARE | / | LfaN-Hall, HalOmi | Sentence | NMT | COMET-QE, LASER, XNLI and LaBSE | Aggregate hallucination scores | AUROC, FPR | ✓ | ✗ |
| | HaluAgent | / | HaluEval-QA, WebQA, Ape210K, HumanEval, WordCnt. | Response, Sent | Knowledge-based QA, Math, Code generation, Conditional text generation. | Baichuan2-Chat, GPT-4 | Sentence Segmentation, Tool Selection and Verification, Reflection | Acc, P, R, F1 | ✓ | ✓ |
| | RefChecker | KnowHalBench | Natural Questions, MS MARCO, databricks -dolly15k | Claim-triplet | Closed-Book QA, RAG, Summ, Closed QA Information Extraction | Mistral-7B, GPT-4, NLI | Extractor and Checker | Acc, P, R, F1 | ✓ | ✓ |
| | HDM-2 | HDMBENCH | RAGTruth, enterprise support tickets, MS Marco, SQuAD, Red Pajama v2. | Word, Response | Generation | Qwen-2.5-3B-Instruct | Classification | P, R, F1 | ✓ | ✓ |
| | Lookback Lens | / | CNN/DM, XSum, Natural Questions, MT-Bench | Response | Summ, QA, Multi-turn conversation | LLaMA-2-7B-Chat, GPT-based | Attention Map | AUROC, EM | ✓ | ✓ |

Table 5: AHE Meta-Info Table after LLM era (Part 1), which means the methods utilize the ability of LLMs such as ChatGPT.

| Era | Name | New Dataset | Data Source | Fact Definition | Task | Based-model | Method | Metric | SF | WF |
|---|---|---|---|---|---|---|---|---|---|---|
| After LLM Era | KnowHalu | / | HaluEval, HotpotQA, CNN/DM | Response | QA, Summ | Starling-7B, GPT-3.5 | Identify non-fabrication, multi-form fact-checking | TPR, TNR, Avg Acc | ✓ | ✓ |
| | AXCEL | / | SummEval, QAGS | claim | Summ, Generation, Data2text | Llama-3-8B,Claude-Haiku, Claude-Sonnet | Direct Assessment | P, R, F1, Auc | ✓ | ✓ |
| | Drowzee | Drowzee | / | Response | QA | GPT-3.5-turbo, GPT-4, Llama2-7B,70B, Mistral-7B-v0.2,8x7B | Direct Assessment | FCH Ratio | ✗ | ✓ |
| | MIND | HELM | / | Span | Continual writing | MLP | Embedding MLP classification | AUC, Pearson corr | ✗ | ✓ |
| | BTProp | / | Wikibio-GPT3, FELM-Science, FactCheckGPT | Response | Generation | gpt-3.5 -turbo, Llama-3-8B Instruct | hidden Markov tree | AUROC, AUC-PR, F1, Acc | ✗ | ✓ |
| | FAVA | FAVABENCH | Open prompts | Span | Information retrieving | Llama2-Chat 7B | Hallucination tags generation | F1 | ✗ | ✓ |
| | Semantic Entropy | / | BioASQ, TriviaQA, NQ Open, SQuAD | Response | QA | LLaMA 2 Chat-7B,13B,70B, Falcon Instruct-7B,40B, Mistral Instruct-7B | Semantic Entropy | AUROC, AURAC | ✗ | ✓ |
| | SEPs | / | BioASQ, TriviaQA, NQ Open, SQuAD | Response | QA | Llama-2-7B,70B, Mistral-7B, Phi-3-3.8B | Semantic Entropy Probes | AUROC | ✗ | ✓ |
| | HaloScope | / | TruthfulQA, TriviaQA, CoQA, TydiQA-GP | Response | QA | LLaMA-2-chat-7B,13B, OPT6.7B,13B | Unsupervised learning | AUROC, BLUERT, ROUGE | ✗ | ✓ |
| | LRP4RAG | / | RAGTruth | Response | QA | Llama-2-7B/13B-chat | Internal state classification | Acc, P, R, F1 | ✓ | ✓ |
| | Halu-J | ME-FEVER | FEVER | Claim | Fact-checking | GPT-4, Mistral-7B-Instruct | Reasoning | Acc | ✗ | ✓ |
| | NonFactS | NonFactS | CNN/DM | Word | Summa | BART-base, ROBERTa ALBERT | NLI | Balanced Acc | ✓ | ✗ |
| | MFMA | / | CNN/DM, XSum | Span, Ent | Summ | BART-base, T5-small, Electra-base-discriminator | Classification | F1, Balanced Acc | ✓ | ✗ |
| | HADEMIF | / | / | Response | QA | Llama2-7B | Expected Calibration Error, Brier Score | acc@q, cov@p | ✗ | ✓ |
| | REDEEP | / | RAGTruth, Dolly (AC) | Span | RAG | Llama2-7B/13B/70B, Llama3-8B | External Context Score, Parametric Knowledge Score | AUC, PCC, Acc, R, F1 | ✗ | ✓ |
| | LMvLM | / | LAMA, TriviaQA, NQ, PopQA | Response | QA | ChatGPT, text-davinci-003, Llama-7B | LMs multi-turn judge | P, R, F1 | ✗ | ✓ |
| | OnionEval | OnionEval | / | Ent, Atomic fact | QA | SLLMs (Llama, Qwen, Gemma) | Layered Evaluation | Acc, Context-influence Score | ✓ | ✗ |
| | LongEval | Guidelines | SQuALITY, PubMed | Response | Summ | LLMs | LLM-as-a-Judge | Score Aggregation | ✓ | ✗ |
| | Adams et al. (2023) | / | MIMIC-III | Response | Summ | Multiple | Meta-evaluation | Pearson | ✓ | ✗ |
| | Jia et al. (2023) | / | SummEval, FRANK, QAGS-XSUM | Response | Summ | Foundation Models | LLM-as-judge (QA) | Pearson | ✓ | ✗ |
| | ACUEval | ACU-Annotations | SummEval, AggreFact, LLMSummEval | ACUs | Summ | GPT-4 | ACU Extraction & Verification | P, R, F1 | ✓ | ✗ |
| | FENICE | / | AggreFact | Claim | Summ | DeBERTa-v3 | Claim Extraction & NLI | P, R, F1 | ✓ | ✗ |
| | Zhang et al. (2024a) | DiverSumm | AggreFact, SummEval, etc. | Sent | Summ | DeBERTa-v3 | Fine-grained NLI | P, R, F1 | ✓ | ✗ |
| | HGOT | / | TruthfulQA, HaluEval | Response | RAG | Llama2 | RAG + Graph of Thoughts | Acc | ✗ | ✓ |
| | ReEval | / | HaluEval, RAG-Benchmark | Response | RAG | Llama2 | Adversarial Attack | ASR | ✓ | ✗ |
| | TimeChara | TimeChara | Literary works | Response | Role-playing | GPT-3.5/4 | LLM-as-a-Judge | Consistency Score | ✗ | ✓ |
| | MetaCheckGPT | / | SemEval-2024 Task 6 | Response | Multi-task | GPT-3.5 | Uncertainty + Meta-model | F1, Acc | ✓ | ✓ |
| | Liu et al. (2024) | Taxonomy | Human-written code | Code Snippet | Code Gen | GPT-4, Code Llama | LLM-as-a-Judge | Hallucination Rate | ✗ | ✓ |
| | Magesh et al. (2024a) | Legal Qs (192) | Lexis+ AI, Westlaw AI, Ask Practical Law AI | Response | Legal QA | GPT-4, Claude | Human Evaluation | Accuracy | ✗ | ✓ |
| | Hou et al. (2024a) | Annotations | CLERC benchmark | Span | Legal Analysis | GPT-4 | Fine-grained Eval | P, R, F1 | ✓ | ✗ |
| | OpenFactCheck | / | FEVER, TruthfulQA, HaluEval | Atomic Fact | Fact Check | GPT-4, Llama | Fact Decomp. & Verification | F1, Acc | ✗ | ✓ |
| | PlainQAFact | PlainFact | Biomedical texts | Response | QA, Summ | GPT-4 | Direct Assessment | Correlation | ✓ | ✗ |
| | Xiong et al. (2025) | TruthHypo | Biomedical Lit | Response | Hypothesis Gen | GPT-4, Gemini | LLM-as-a-Judge | Truthfulness Score | ✗ | ✓ |
| | MedScore | AskDocsAI | Reddit (r/AskDocs), PUMA | Response | QA | GPT-4 | Statement Verification | MedScore | ✗ | ✓ |
| | T2F | / | Unstructured text | Fact Items | Factuality Eval | LLM-agents | Multi-agent Framework | P, R, F1 | ✓ | ✗ |
| | VeriFact | FactRBench | Long-form text | Triplet | Summ | GPT-4 | Fact Extraction & Verification | P, R, F1 | ✓ | ✗ |
| | VeriFastScore | / | TruthfulQA, Self-generated | Sent | Summ | DeBERTa-v3 | NLI-based Alignment | Pearson | ✓ | ✗ |
| | Luna | / | Real-world RAG data | Sent | QA | DeBERTa-v3 | NLI (3-class) | F1, Acc | ✓ | ✗ |
| | Jing et al. (2025) | / | ConvoAS, ConvoTS, ReviewTS, JsonTG | Response | Summ | GPT-4 | Likert Scale Eval | Pearson | ✓ | ✗ |
| | Dong et al. (2025) | / | Multi-round reasoning tasks | Response | Reasoning | GPT-4 | LLM-as-a-Judge | Accuracy | ✗ | ✓ |
| | Wang et al. (2025) | / | GSM8K, MATH, StrategyQA | Response | Reasoning | LLMs | Answer & Reasoning Consistency | Acc, F1 | ✗ | ✓ |
| | Cheng et al. (2025) | / | TruthfulQA, TriviaQA, NQ | Response | QA, Reasoning | Llama2 | Semantic Entropy | AUC, Acc | ✗ | ✓ |

Table 6: AHE Meta-Info Table after LLM era (Part 2), which means the methods rely on the ability of LLMs such as ChatGPT.

## C.2 Dataset and Benchmark URLs

This section provides a list of the full URLs for all datasets and benchmarks discussed in our survey. As detailed in Table 7, this compilation is intended to serve as a practical reference to facilitate reader access and support further research.

| Dataset Name | Full URL |
|---|---|
| DialogueNLI | https://wellecks.com/dialogue_nli/ |
| CoGenSumm | https://tudatalib.ulb.tu-darmstadt.de/items/ 9a3612a3-4fba-400f-8b23-bf1e917d894f |

Continued on next page

Table 7 – continued from previous page

| Dataset Name | Full URL |
| --- | --- |
| XSumFaith | https://github.com/google-research-datasets/xsum_hallucination_annotations |
| QAGS | https://github.com/W4ngatang/qags |
| Polytope | https://github.com/hddbang/PolyTope |
| FRANK | https://github.com/artidoro/frank |
| Falsesum | https://github.com/joshuabambrick/Falsesum |
| FactEval | https://github.com/BinWang28/FacEval |
| Devaraj et al. (2022) | https://github.com/AshOlogn/Evaluating-Factuality-in-Text-Simplification |
| NonFactS | https://github.com/ASoleimaniB/NonFactS |
| RefMatters | https://github.com/kite99520/DialSummFactCorr |
| DiaHalu | https://github.com/ECNU-ICALK/DiaHalu |
| TofuEval | https://github.com/amazon-science/tofueval |
| RAGTruth | https://github.com/ParticleMedia/RAGTruth |
| SummaCoz | https://huggingface.co/datasets/nkwbtb/SummaCoz |
| FaithBench | https://github.com/vectara/FaithBench |
| Q2 | https://github.com/orhonovich/q-squared/tree/main |
| HADEs | https://github.com/yizhe-zhang/HADES |
| TruthfulQA | https://github.com/sylinrl/TruthfulQA |
| FACTOR | https://github.com/AI21Labs/factor |
| HaluEval | https://github.com/RUCAIBox/HaluEval |
| PHD | https://github.com/maybenotime/PHD |
| FAVA | https://fine-grained-hallucination.github.io/ |
| THaMES | https://github.com/holistic-ai/THaMES |
| HELM (MIND) | https://github.com/oneal2000/MIND/tree/main |
| HalluLens | https://github.com/facebookresearch/HalluLens |
| FreshLLMs | https://github.com/freshllms/freshqa |
| ERBench | https://github.com/DILAB-KAIST/ERBench |
| KOLA | https://github.com/thu-keg/kola |
| RealtimeQA | https://github.com/realtimeqa/realtimeqa_public |
| FactBench | https://github.com/f-bayat/FactBench |
| FactScore | https://github.com/shmsw25/FActScore |
| BAMBOO | https://github.com/RUCAIBox/BAMBOO |
| ChineseFactEval | https://gair-nlp.github.io/ChineseFactEval/ |
| HalluQA | https://github.com/OpenMOSS/HalluQAEval |
| UHGEval | https://github.com/IAAR-Shanghai/UHGEval |
| ANAH | https://github.com/open-compass/ANAH |
| HalOmi | https://github.com/facebookresearch/stopes/tree/main/demo/halomi |
| Chinese SimpleQA | https://github.com/he-yancheng/Chinese-SimpleQA |
| C-FAITH | https://github.com/PKU-YuanGroup/C-FAITH |
| Bi'an | https://github.com/NJUNLP/Bian |
| HalluVerse25 | https://doi.org/10.48550/ARXIV.2503.07833 |
| Poly-FEVER | https://doi.org/10.48550/ARXIV.2503.16541 |
| MASSIVE | https://github.com/alexa/massive |
| MultiHal | https://doi.org/10.48550/ARXIV.2505.14101 |
| K-HALU | https://github.com/jaehyung-seo/k-halu |
| MedHalt | https://medhalt.github.io/ |
| MedHallu | https://medhallu.github.io/ |
| LegalHallu | https://huggingface.co/datasets/reglab/legal_hallucinations |
| SUMMEDITS | https://huggingface.co/datasets/Salesforce/summeditsl |
| DefAn | https://github.com/saeed-anwar/DefAn-Benchmark |
| HalluMix | https://github.com/deanna-emery/HalluMix |
| ToolBeHonest | https://github.com/ToolBeHonest/ToolBeHonest |
| RoleBench | https://doi.org/10.48550/ARXIV.2411.07965 |
| Molecular Mirage | https://github.com/H-ovi/Molecular-Mirage |
| Collu-Bench | https://github.com/collu-bench/collu-bench |
| TIB (Traffic Incident Benchmark) | https://doi.org/10.18653/v1/2025.naacl-industry.4 |
| Wizard of Wikipedia | https://parl.ai/projects/wizard_of_wikipedia/ |

Table 7 – continued from previous page

| Dataset Name | Full URL |
|---|---|
| TopicalChat | `https://github.com/alexa/Topical-Chat` |
| SummEval | `https://github.com/Yale-LILY/SummEval` |
| BEAMetrics | `https://github.com/ThomasScialom/BEAMetrics` |
| CI-ToD | `https://github.com/yizhen20133868/CI-ToD` |
| SummaC | `https://github.com/tingofurro/summac` |
| BEGIN | `https://github.com/google/BEGIN-dataset` |
| FaithDial | `https://huggingface.co/datasets/McGill-NLP/FaithDial` |
| DialSumMeval | `https://github.com/kite99520/DialSummEval` |
| TRUE | `https://github.com/google-research/true` |
| AGGREFACT | `https://huggingface.co/datasets/lytang/LLM-AggreFact` |
| FELM | `https://github.com/hkust-nlp/felm` |

Table 7: List of Dataset URLs for Reference.

