# OpenReview forum: "A Survey of Automatic Hallucination Evaluation on Natural Language Generation"
_TMLR — Withdrawn by Authors_

### Review · Reviewer_tJyT · 2025-08-11

**Summary Of Contributions:**

This paper presents a survey (a compilation) of existing work on hallucination detection. The authors organise existing work on three axes: datasets, type of evidence, and detection method. Each axe further organises into sub-categories. The dataset axe, discusses existing hallucination evaluation datasets in different tasks, e.g., summarisation, question answering, and newer challenging tasks such as long-context understanding or multi-lingual settings. The type of evidence axe defines two specific hallucinations subcategories, namely Source Faithfulness (SF) when the evaluation depends on a given evidence, and World Faithfulness (WF) when the assessment is w.r.t. evidence that needs to be found/retrieved. The methods axe, describes different approaches to hallucination detection (token, semantic, or LLM-based).

**Audience:**

No

**Claims And Evidence:**

No

**Requested Changes:**

- The organisation and description of the survey needs to be reformulated. The survey could perhaps be organised according to usability aspects of hallucination detection approaches, e.g., latency, requirements for supervision (training data and/or fine-tuning), correlation with human judgements, interpretability, robustness, generalisability, multi-linguality, required evidence, required reference, granularity of the evaluation, usage of additional modules or not (e.g., question generation),  etc. Are older benchmarks and methods still useful in the context of current LLM generators? Are hallucinations in current models the same as in older ones? Perhaps those aspects enumerated in Section 6 (future directions) could be used to organise the discussion of existing approaches. Or maybe start by enumerating the key desired features of today hallucination detection methods, and from there work out all methods (including old ones) discussing what do they provide and how.

- Saying that the AHE methods follow a pipeline seems confusing, what the authors call a pipeline seems confusing. On one hand, there exist work on benchmarks/datasets for meta-evaluation of hallucination evaluators. On the other hand, it is the working of the approaches, the type of evidence and the method itself.

- (Intro) It is not clear that old NLG approaches overlooked hallucinations. Although the terminology was not introduced they did evaluate faithfulness and factuality in the generated text (e.g. [1,2]). Many old papers look at the issue though not calling it hallucination, not only (Lee et al., 2018). For instance,

[1] Bootstrapping Generators from Noisy Data (human evaluation of faithfulness in data-to-text)
https://aclanthology.org/N18-1137/

[2] Get To The Point: Summarization with Pointer-Generator Networks (an architecture to improve factual errors in summarisation)
https://aclanthology.org/P17-1099/


- Section 5.2, Table 2. It would be useful to recall what type is each evaluator, i.e., SF / WF / both. It would be useful to have results on the entire dataset.

- The authors should properly format references, including for each reference full details about the type/place/etc. of its publication.



- Missing references:

Language Models (Mostly) Know What They Know (LLm-based, self-consistency)
https://arxiv.org/abs/2207.05221

Leveraging Entailment Judgements in Cross-Lingual Summarisation (multi-lingual, hallucination evaluation cross-language)
https://aclanthology.org/2024.findings-acl.862/

Fine-grained natural language inference based faithfulness evaluation for diverse summarisation tasks (NLI, SF based, finer granularity)
https://aclanthology.org/2024.eacl-long.102/

The Hallucinations Leaderboard--An Open Effort to Measure Hallucinations in Large Language Models (Emerging Hallucination Types, LLMs/Tasks leaderboard)
https://arxiv.org/pdf/2404.05904?

Uncertainty Quantification in Retrieval Augmented Question Answering (hallucinations in RAG QA via uncertainty)
https://arxiv.org/pdf/2502.18108?

**Strengths And Weaknesses:**

### Strengths

- A comparison and/or evaluation of the many existing approaches to hallucination evaluation could be useful.

### Weaknesses

- The categorisation w.r.t. the three axes does not seem to add much value to the survey beyond the compilation of existing approaches.

- The survey does not seem to have a good balance of details. On each section/categorisation, it enumerates many papers, giving one sentence for each paper with too general paper details which do not help in understanding it. See example [a] below.

[a] "Drowzee (Li et al., 2024b) detects fact-conflicting hallucinations by applying logic-reasoning-based data mutation through five custom-designed rules and deploying two semantic-aware oracles".
What are "fact-conflicting hallucinations"?
What is "logic-reasoning-based data mutation"?
What are the "five custom-designed rules"?
How does this approach work?

- The authors discuss categories such us binary or fine-grained hallucination labels (enumerating existing methods); but beyond enumerating them, there seems to be no overall knowledge or insight w.r.t. to these hallucination label variants.

- Not clear what the authors meant and how the survey addresses the claim in the abstract "the field suffers from methodological
fragmentation, hindering both theoretical understanding and practical advancement".

- "we identify fundamental limitations in current approaches and their implications for real-world deployment". The qualitative analysis (Section 5.2, Table 2) seems limited to support this claim.

- The survey does not include faithfulness in the context of data-to-text.

Handling Divergent Reference Texts when Evaluating Table-to-Text Generation
https://arxiv.org/pdf/1906.01081

---

> ### Author Response · Authors · 2025-08-25
>
> Dear Reviewer tJyT,
>
> Thank you for your thoughtful and constructive feedback on our manuscript!
>
> We are grateful for your detailed analysis, which has helped us to significantly improve the structure, depth, and overall contribution of our survey.
>
> In response, we have undertaken a major revision of the paper. The most significant change is that we adjust the detail level of the methods and reframe the discussion section. We have also incorporated the missing literature and clarified our terminology.
>
> Below, we address each of your comments in detail, explaining the specific changes we have made:
>
> ---
>
> > The organisation and description of the survey needs to be reformulated. …
> >
>
> **Response:** We believe this is the most critical point of your review, and we thank you for this invaluable suggestion. We agree that the value of a survey lies not just in categorization, but also in answering these deeper questions.
>
> We have added some sections to the paper to be more analytical. We have introduced a deeper discussion section, which is now organized around the usability aspects and critical questions. This section serves as the analytical core of the paper and contains: **Key Dimensions of AHE, The Evolving Landscape of Hallucination and Evaluation,** and **The Complexities of Hallucination in Real-World Applications.**
>
> By reframing this analytical discussion, we now directly use our taxonomy to provide insights into the field's progress, limitations, and trajectory, rather than simply enumerating papers.
>
> > The survey does not seem to have a good balance of details…
> >
>
> **Response:** In our revised survey, which now covers 105 papers (up from 74), providing an in-depth analysis of every single method is impractical. Therefore, we now balance detail by grouping and comparing similar methods, while also providing more substantial descriptions for the most representative works. More importantly, the new Discussion section allows us to provide a deeper understanding of their collective contributions and limitations.
>
> ---
>
> > Not clear what the authors meant and how the survey addresses the claim in the abstract ‘the field suffers from methodological fragmentation, hindering both theoretical understanding and practical advancement’.
> >
>
> **Response:** Thank you for highlighting this. We argue that the "methodological fragmentation" is directly illustrated by the taxonomy we present in Section 3, Figure 2 and Figure 3. The sheer diversity of approaches, spanning reference-based, reference-free, and LLM-based paradigms, each with numerous sub-techniques, demonstrates a lack of a unified evaluation standard. Our survey systematizes this fragmented landscape, which is the first step toward addressing it.
>
> ---
>
> > The survey does not include faithfulness in the context of data-to-text.
> >
>
> **Response:** We have now included the context of data-to-text generation in the Introduction and Section 3.1.1 & 3.1.2.
>
> ---
>
> > (Intro) It is not clear that old NLG approaches overlooked hallucinations…
> >
>
> **Response:** We have revised the introduction to reflect this, as highlighted in yellow.
>
> ---
>
> > Section 5.2, Table 2. It would be useful to recall what type is each evaluator…
> >
>
> **Response:** We have now clarified SF and WF again for this table. The case study was provided primarily to illustrate how the same evaluator behaves when encountering different types of hallucinations.
>
> Specifically, we were interested in the rare but critical scenario where a summary is faithful to a factually incorrect source. These cases are uncommon in standard summarization datasets (where sources are typically factual) and are more characteristic of tasks like false-premise QA. A full-dataset quantitative analysis on these summarization benchmarks would not have provided sufficient examples to draw meaningful conclusions about this specific failure mode. For better coherence of the main text, we have moved this case study to the Appendix, where interested readers can find further details.
>
> ---
>
> > References format
> >
>
> **Response:** Thank you for noting this. We have carefully reviewed and corrected the entire bibliography to ensure that all references are complete.
>
> ---
>
> > Missing references
> >
>
> **Response:** We are very grateful for this list of important and relevant work. We have incorporated all of your suggested references into our manuscript and its discussion:
>
> - Section 1: The Hallucinations Leaderboard
> - Section 2.3: Leveraging Entailment Judgements in Cross-Lingual Summarisation
> - Section 3.1 NLI: Fine-grained natural language inference based faithfulness evaluation for diverse summarisation tasks
> - Section 3.2, uncertainty: Language Models (Mostly) Know What They Know; Uncertainty Quantification in Retrieval Augmented Question Answering
>
> ---
>
> We hope these revisions fully address your concerns. We sincerely thank you once again for your constructive guidance.
>
> Best regards,
>
> Authors of Paper5135

---

> ### Comment · Reviewer_tJyT · 2025-09-01
>
> Thank you to the authors for the detailed response. My main concern about the focus and contribution of the survey is not fully addressed. Improving this requires major modifications and rethinking of the current presentation (which seems an overview of every possible sub-topic related to hallucination evaluation). I agree with reviewers HGJ3 and Kczb and suggest the authors to take the time to make this a useful survey.

---

### Review · Reviewer_Kczb · 2025-08-15

**Summary Of Contributions:**

The authors divide work on Automatic Hallucination Evaluation (AHE) into three broad categories: datasets & benchmarks, evidence collection, and comparison and judgement. Surveying 74 recent papers, the authors also distinguish between "Source Faithfulness" and "World Factuality" as they dig deeper into the taxonomy, ultimately arriving at 29 different kinds of contributions in the space of AHE. This listing might be helpful to someone looking for a jumping off point into the recent literature, though see also Weaknesses below.

**Audience:**

Yes

**Broader Impact Concerns:**

No broader impact concerns.

**Claims And Evidence:**

No

**Requested Changes:**

## Most Essential Feedback
- Please systematize your inclusion criteria for papers in your review and expand the pool of papers if necessary. The effort to develop a general taxonomy is undercut by the fact that the papers seem to be a convenience sample of work on this topic rather than a well-defined survey of the field. One useful resource for generating an overview of your selection process is https://www.eshackathon.org/software/PRISMA2020.html , which also has useful links to the PRISMA guidelines which are themeselves useful for developing a structured survey.

## Additional Feedback (some minor, some more substantial)
### Intro
- the "like an elephant in the room" sentence implies that input and world accuracy have long been issues within NLG, but early NLG systems were oriented toward expressing all and only the content of the input, often using rule-based systems that guaranteed factual accuracy but resulted in stilted or awkward phrasing. I don't think this is an accurate representation of the history of the field.
- "AHE proves crucial..." -- proofs and proving things are rare in science; for this colloquial usage, it might be sensible to replace "proves" with "will be", although there's still no evidence or argumentation presented to establish that AHE is necessary for LLM safety and reliability.
- paragraph 2 -- the phenomenon commonly appeared in work on data-to-text generation using NNs as well.
- -- which NLG tasks became achievable?
- -- their responses *frequently* contain hallucinations
- Figure 1 -- the comment about Kelce in the caption does not match the image
- WF by definition presents more expansive challenges: indeed, it is fundamentally impossible to assess the "world factuality" of any and all statements, whether made by humans or sampled from an LLM. I think the SF vs. WF distinction is fundamentally about whether the answer should be aligned with something from the input or with something external to the input.
- final paragraph before 1.1 -- "hallucinations exhibit greater diversity" -- greater compared to what?
- the final sentence of this paragraph could go as the first sentence of 1.1
- Figure 2 -- Please double check the colors of your figure for grayscale and colorblindness appropriateness or adjust the styling so that the color groups are visually distinguishable without relying on color (e.g. relying on shape or texture). It is not immediately clear that the middle column of groups (Evidence and Text to be Evaluated) are a different kind of thing than the Data & Evidence or the Comparison sections. Boldface is not used consistently for section labels and the dotted lines cross the headers for several groups.
- "nor do they systematically summarized evaluator processes" -- grammar
- "up-to-date" -- what date? let us know when the survey was conducted
### Dataset & Benchmark
- "Of the evaluators", which evaluators? It's not clear who this is referring to.
- S2.1 -- "binary classification... often lacks granularity" -- yes, by definition. it only has two categories. what is the important point here?
- For Table 1 and others, the choice of font clashes with the others used in the paper, and the font size is quite small, even when enlarged on a screen. You may need to rotate the tables 90 degrees to fit more nicely on the page at a larger font size. I would also advice including the full text of the links in the appendix or the references, so that it is still possible to enter the links manually from a printed copy of the paper or when using a PDF viewer which does not handle links well.
### Evidence collection
- "Datasets and benchmarks provide the foundation for AHE" is immediately contradicted by the rest of this section, which includes efforts to evaluate hallucinations without reference to existing datasets or benchmarks.
- "This step is grounded in the premise..." -- this sentence is unclear
- "a more practical alternative is" --> "a more practical alternative may be"
- "Large scale automation in evidence gathering..." -- this sentence is a strong claim!
- "In this section we focus on evidence collection strategies..." -- If this is the intended distinction from datasets & benchmarks, maybe the section headings should emphasize the difference between AHE grounded in existing gold standard data versus AHE based on searching for evidence?
- "Entire input as Evidence" -- the last sentence says there are significant flaws but does not name them
- S3.2 -- external KBs cannot offer a comprehensive reservoir of world knowledge, because this is an impossible task. However, there are large KBs that capture a lot of what people know about the world, but I don't think they are *comprehensive*
- "Knowledge Base" is usually used to refer to structured data, not collections of documents, so it is important to make clear that you are using an expansive definition which includes things like Wikipedia and PubMed and not just structured knowledge bases.
- LLM as KB -- citations are needed to establish that LLMs have "learned knowledge while training" and "can serve as KBs"
- what is the distinction between "by incorporating information retrieved at inference time" and "online search" or otherwise accessing another knowledge base?
- Is "parametric knowledge" a generally accepted term meaning "the knowledge encoded in a neural network's parameters"? I am unfamiliar with this as a fixed expression, so do consider defining it in your paper, especially as survey papers frequently attract newer researchers.
- S3.3 -- what is the difference between a "unified framework" and a "joint evaluation"?
### Comparison and Judge
- Indeed n-gram overlaps are poor measures of text quality or accuracy in general; however they offer more flexibility than Exact Match scores, which are included in the survey, so I am unsure why they are completely excluded from discussion.
- QG-QA -- I believe this term was defined earlier in the paper, but it's worth writing out what the acronym stands for here to help refresh the reader's memory
- QA Benchmark -- how is accuracy measured relative to the "ground-truth answers for evaluation"?
- S4.2 -- lexica include more than just a list of words -- they also document their morphological properties, semantics, and relationships to other words. Consider reframing 4.1 as having to do with string matching and 4.2 as having to do with inference and entailment based evaluations.
- It is unclear why "Data Augmentation in NLI" is relevant to this survey and the subsequent two sections don't feel like they fit in S4.2.
- S4.3 -- this section represents a small cross-section of LLM-as-a-Judge work, while also not limiting the works explicitly to AHE
- S4.4 -- a model can be fully self-consistent while hallucinating, so long as it hallucinates the same thing in the same context -- this issue seems orthogonal to the problem under discussion
- S4.5 -- where is the evidence that it is "straightforward" to use an LLM for evaluation? This does not seem to follow from the survey so far.
### Discussion
- "many challenges have been addressed or mitigated" -- have they? which ones? and how?
- what does it mean to be "detached from the input"
- why should we assume there is an optimal granularity level in general?
- "Is hallucination always bad?" is a strange section that seems to stretch the definition of hallucination or violate it altogether. In the legal example, incorporating accurate external knowledge would only be a hallucination according to your SF analysis, but not according to WF. Similarly, is the concept of "accuracy" well-defined for creative tasks? I don't think this section contributes to the overall paper in its current form.
- S5.2 -- what is the intended role of these examples with respect to the rest of the survey? this section might fit better in the introduction when you are defining SF and WF to provide more examples for the reader, though then it would be beneficial to carry the discussion of SF vs. WF on through the rest of the survey and taxonomy.
- S5.3 -- what do you mean by "the dialectical relationship" between SF and WF?
- S5.4 -- it would be good to cite related surveys of human evaluation as points of reference here
- "Standardized Although" --> "Although"
- S6, Efficiency -- "without invoking full LLM inference will" -- change "will" to "would". Similarly, for the "will"s in the second and fourth claims, how do you know these will be the outcomes? Please express a degree of certainty commensurate with the evidence.
- Some folks are already doing "hybrid annotation" strategies, calling them active learning or human-in-the-loop or by other names, so this is not a novel proposal.
- "Emerging Hallucination Types" -- these all seem to be subtypes of SF and WF.
### Conclusion
- "directly affects the reliability, safety, and overall applicability...across diverse tasks and models" -- how does evaluating hallucination affect these things?
- "Collaboration...using...linguistics...cognitive science, and human evaluation" -- of these you have only discussed human evaluation. Why are the others being mentioned here? I agree they are relevant to improving AHE, but this paper has not argued that.

**Strengths And Weaknesses:**

## Strengths:
- Hallucination (i.e. inserting information not present in the input for input-based tasks or factually inaccurate information for general-purpose / sampling-only tasks) is a critical problem for LLMs as a core language technology, making the topic very timely.

## Weaknesses:
- This survey is not systematic and reproducible, limiting the utility of the taxonomy and the findings.
- Key facts relevant for the practitioner seeking to identify promising approaches to AHE are missing from the survey, such as performance information for different AHE methods.

---

> ### Author Response · Authors · 2025-08-25
>
> Dear Reviewer Kczb,
>
> Thank you for your incredibly thorough and insightful review of our manuscript! They have greatly helped us refine our work and improve its quality, rigor, and clarity.
>
> In particular, following your recommendation, we revised our methodology into a database-driven literature survey guided by the PRISMA workflow, with a detailed flowchart included in the Appendix. We are sincerely grateful for your guidance in pointing us toward the PRISMA framework website!
>
> We carried out a structured literature search on the DBLP database covering the past five years. We are glad to have now combined our previously searched papers with the DBLP-based approach for a more comprehensive survey.
>
> We have carefully addressed every point you raised and have performed a substantial revision of the manuscript. To make our responses clearer, we present a detailed point-by-point reply to your comments below.
>
> ### **General Weaknesses**
>
> > Key facts relevant for the practitioner seeking to identify promising approaches to AHE are missing from the survey, such as performance information for different AHE methods.
> >
>
> **Response:**
> This is a valid point. While a direct, normalized performance comparison is challenging due to varied experimental settings, we have added extensive meta-information to aid practitioners. Tables 3, 4, and 5 in Appendix B now provide detailed meta-information for all 105 surveyed methods. These tables outline the underlying models, core methodology, and evaluation metrics for each approach, allowing readers to compare them based on their technical characteristics.
>
> ---
>
> ### **Introduction**
>
> > Data-to-text generation should be mentioned as a task where hallucinations appeared.
> >
>
> **Response:**
> We have added more related papers in the Introduction and Section 3.1.2 & 3.1.2.
>
> > Figure 2’s color and styling.
> >
>
> **Response:**
> We appreciate this feedback on accessibility. We have adjusted the category layer to have sharp corners and updated the caption.
>
> **Other comments:**
>
> > -  the "like an elephant in the room" sentence implies that …
> > - "AHE proves crucial" is too strong; "will be" is better. The necessity of AHE is not argued.
> > - The caption for Figure 1 regarding Travis Kelce does not match the image.
> > - which NLG tasks became achievable?
> > - their responses *frequently* contain hallucinations
> > - "hallucinations exhibit greater diversity"—greater compared to what?
> > - "nor do they systematically summarized evaluator processes" -- grammar
> > - "up-to-date"—what is the cutoff date for the survey?
> >
>
> **Response:**
> Thank you for these detailed suggestions. We have modified the text to address each point or made the information explicit. Changes are highlighted in yellow in the revised manuscript.
>
> ---
>
> ### **Dataset & Benchmark**
>
> > "Of the evaluators"—this reference is unclear.
> >
>
> **Response:**
> We have made it explicit, highlighted in yellow.
>
> > S2.1 - The point that "binary classification... often lacks granularity" is obvious by definition.
> >
>
> **Response:**
> Our intention was to use it as a transition to introduce the field's evolution toward more sophisticated, fine-grained benchmarks. We have rephrased the text in Section 2.1 to better serve this transitional purpose.
>
> > Issues with table formatting (font, size) and inaccessible links from a printed copy.
> >
>
> **Response:**
> We have split the table across two pages and increased the font size, as the datasets have also been expanded. In addition, we provide the full link in the Appendix.
>
> ---
>
> ### **Methodology (Evidence)**
>
> > The opening sentence, "Datasets and benchmarks provide the foundation for AHE," is contradicted by the discussion of reference-free methods.
> >
>
> **Response:**
> Thank you for highlighting this logical inconsistency. We have performed a restructuring of the paper. Section 2 is now dedicated exclusively to datasets and benchmarks, while Section 3 independently covers AHE methodologies, including reference-free approaches. This parallel structure resolves the contradiction.
>
> > The section on "Entire input as Evidence" mentions significant flaws but does not name them.
> >
>
> **Response:**
> Thank you for catching this omission. This part is rephrased in Section 3.1.1 SF Evidence as “introduces significant noise” and “information redundancy”.
>
> > The definition of "Knowledge Base" is used expansively and should be clarified.
> >
>
> **Response:**
> This is an important point of clarification. We now make it clear in Section 3.1.1 that we use an expansive definition of "Knowledge Base" to include both structured databases (e.g., for ERBench) and large-scale knowledge corpora like Wikipedia and PubMed (e.g., for Med-HALT).
>
> > Citations are needed to support the claim that LLMs can serve as a "KB".Is "parametric knowledge" a generally accepted term? It should be defined.
> >
>
> **Response:**
> In Section 3.1.1, we have added citations and an explanation of “parametric knowledge”.

---

> > ### Author Response · Authors · 2025-08-25
> >
> > > What is the distinction between "incorporating information retrieved at inference time" and "online search"?
> > >
> >
> > **Response:**
> > "Online search" is a particular method of accessing information. In practice, the LLM usually performs this search on its own, automatically filtering and integrating the results.
> >
> > **Others:**
> >
> > > - The sentence "This step is grounded in the premise..." is unclear.
> > > - Suggestion to change "a more practical alternative is" to "...may be".
> > > - The statement "Large scale automation in evidence gathering..." is a very strong claim.
> > > - Section headings should better distinguish between …S3.2 - External KBs are not a "comprehensive" reservoir of world knowledge.
> > > - S3.3 - What is the difference between a "unified framework" and a "joint evaluation"?
> > >
> >
> > **Response:**
> > We appreciate you pointing these out. As part of our major revision of Section 3, the descriptions of methodologies have been rewritten for improved clarity and logical flow, and we have softened overly strong claims. The specific unclear sentences have been removed or rephrased.
> >
> > ---
> >
> > ### **Methodology (Comparison & Judge)**
> >
> > > QA Benchmark -- how is accuracy measured relative to the "ground-truth answers for evaluation"?
> > >
> >
> > **Response:**
> > We clarify in Section 3.1.2 that accuracy is typically measured by comparing the model's output to ground-truth answers using lexical metrics like Exact Match (EM) scores.
> >
> > ---
> >
> > > It is unclear why "Data Augmentation in NLI" is relevant to this survey.
> > >
> >
> > **Response:**
> > We moved the “Data Augmentation” part to Section 2.5, “Automated Dataset Generation.”
> >
> > ---
> >
> > > S4.3 (now 3.3) represents a small cross-section of LLM-as-a-Judge work, while also not limiting the works explicitly to AHE.
> > >
> >
> > **Response:**
> > We acknowledge that the field of LLM-as-a-Judge is vast. Our survey aims to cover the foundational and most representative works applied specifically to hallucination evaluation. With our expanded literature search, Section 3.3 now provides a more comprehensive overview of this paradigm.
> >
> > ---
> >
> > > S4.4 (now 3.2) - A model can be fully self-consistent while hallucinating.
> > >
> >
> > **Response:**
> > We completely agree with this critical point. This limitation is now explicitly acknowledged and discussed in Section 4.1, where we state that its "critical flaw is an inability to detect WF hallucinations where the model may be confidently and consistently wrong."
> >
> > ---
> >
> > > S4.5 (now 3.3) - Evidence is needed to support the claim that it is "straightforward" to use an LLM for evaluation.
> > >
> >
> > **Response:**
> > Thank you for pointing out this imprecise wording. We have revised the language. We now state that using LLMs as evaluators has proven feasible and effective (supported by citations in Section 3.3), but we no longer claim it is "straightforward," as it comes with its own set of challenges.
> >
> > ---
> >
> > > Why are n-gram overlaps excluded from discussion when Exact Match is included?
> > The acronym "QG-QA" should be written out again for the reader's memory.
> > Consider reframing sections to be about "string matching" vs. "inference and entailment".
> > >
> >
> > **Response:**
> > Thank you for pointing these out. In Section 3, we revised the methodology descriptions for clarity and balance, and the unclear sentences have been removed or rephrased.
> >
> > ---
> >
> > ### **Discussion**
> >
> > > "many challenges have been addressed or mitigated"—which ones and how?
> > >
> >
> > **Response:**
> > This is a fair request for specificity. We have revised the opening of the **Discussion (Section 4)** to be more precise. We now clarify that while foundational challenges (e.g., detecting simple factual contradictions in task-specific settings) have seen progress through the methods surveyed, the increasing sophistication of LLMs introduces new, more subtle challenges that the discussion proceeds to analyze.
> >
> > ---
> >
> > > what does it mean to be "detached from the input”
> > >
> >
> > **Response:**
> > Thank you for asking for clarification. This phrase refers to outputs in translation that “can be either oscillatory (i.e. contain erroneous repetitions of words and phrases) or largely fluent.”[1]
> >
> > ---
> >
> > > Why should we assume there is an optimal granularity level for evaluation?
> > >
> >
> > **Response:**
> > We intended to question if there is an optimal granularity level. We have reframed our discussion to argue against this idea in Section 4.1. In it, we explicitly state that "the assumption of a single, universally optimal granularity is a misconception" and that the ideal level is intrinsically tied to the application's specific goals and error tolerance.
> >
> > ---
> >
> > > The section "Is hallucination always bad?" is strange and stretches the definition.
> > >
> >
> > **Response:**
> > We have replaced that section with a more nuanced discussion in Section 4.3. This section now argues that not all deviations are errors and we are trying to distinguish between unintentional hallucinations and desirable, task-appropriate behaviors like knowledge-informed abstraction in legal summaries or imagination in creative writing.
> >
> > ---

---

> > > ### Author Response · Authors · 2025-08-25
> > >
> > > > The role of examples in S5.2 is unclear and might fit better in the introduction.
> > > >
> > >
> > > **Response:**
> > >
> > > The case study was included primarily to illustrate how the same evaluator behaves when facing different types of hallucinations. In the introduction, Figure 1 already provides illustrative examples of SF and WF. To maintain better coherence in the main text, we have moved this case study to the Appendix, where interested readers can find additional details.
> > >
> > > ---
> > >
> > > > What is meant by the "dialectical relationship" between SF and WF?
> > > It would be good to cite related surveys of human evaluation.
> > > "Standardized Although" is a typo.
> > > >
> > >
> > > **Response:**
> > > Thank you for pointing these out. We have reframed the Discussion section and removed or rephrased the misconceptions and typos, also added essential citations.
> > >
> > > ---
> > >
> > > ### **Future Directions & Conclusion**
> > >
> > > > "Emerging Hallucination Types" all seem to be subtypes of SF and WF.
> > > >
> > >
> > > **Response:**
> > > Our goal was to highlight that these subtypes appear in new, complex domains and present unique challenges. We have clarified in the **Future Directions section on "Emerging Domains"** that these are indeed novel types of hallucination that can be categorized within the SF/WF framework but are distinct because they arise from new contexts (e.g., tool use, cross-modal inconsistencies) and require specialized evaluation frameworks.
> > >
> > > ---
> > >
> > > > How does evaluating hallucination "directly affect reliability, safety, and overall applicability”?
> > > >
> > >
> > > **Response:**
> > > Thank you for asking for this clarification. We have clarified this logic in the **Conclusion (Section 6)**. We now state that robust evaluation is not "merely a diagnostic exercise, but for mitigating risks, advancing model development, and fostering user trust," which are the mechanisms through which reliability and safety are ultimately affected.
> > >
> > > - Reliability: Hallucination evaluation provides a quantitative feedback loop for identifying weaknesses in models and improving their accuracy and dependability.
> > > - Safety: In high-stakes domains such as medicine and law, evaluation is essential for detecting and mitigating risky outputs, ensuring safe deployment.
> > > - Applicability: Rigorous evaluation builds user trust and demonstrates factual accuracy, which is key to real-world adoption.
> > >
> > > ---
> > >
> > > > - Hedging of claims: "will" should be "would" to express appropriate uncertainty.
> > > > - The proposal for "hybrid annotation" is not novel (e.g., active learning, human-in-the-loop).
> > > > - The Conclusion mentions linguistics and cognitive science, which were not discussed in the paper.
> > > >
> > >
> > > **Response:**
> > > We have revised the corresponding paragraph of the conclusion to remove or rewrite these fields. It now mentions fields directly relevant to our survey.
> > >
> > > [1] David Dale, et.al. Detecting and Mitigating Hallucinations in Machine Translation: Model Internal Workings Alone Do Well, Sentence Similarity Even Better. In ACL, pp. 36–50, 2023a. doi: 10.18653/v1/2023.acl-long.3.
> > >
> > > ---
> > >
> > > We hope that these revisions thoroughly address your concerns, and we would like to express our sincere appreciation once again for your insightful and invaluable guidance.
> > >
> > > Best regards,
> > >
> > > Authors of Paper5135

---

> > > > ### Comment · Reviewer_Kczb · 2025-08-31
> > > >
> > > > Thank you for your extensive responses.
> > > >
> > > > Based on past experience, systematic reviews of the literature take time to do well, so I find it somewhat concerning how rapidly new papers are being added to your revised manuscript and we are being assured that they simply fit within the framework previously established. I don't really think this is work that can be done during the discussion process, and that the degree of work required to make this survey comprehensive and sound will take more time and require revision and resubmission, rather than revision for immediate acceptance.
> > > >
> > > > This work is valuable, and I do not want to discourage you from trying to analyse and make sense of this important topic. On the contrary, I want to see this work become the best version of itself so that it can become an invaluable resource for researchers trying to navigate the complexity of the literature today. Good luck with your efforts! I hope to see a future version of this paper published within the next year :)

---

> > > ### Comment · Reviewer_Kczb · 2025-08-31
> > >
> > > **Re "information retrieval vs. online search"** -- what are the other methods of information retrieval that appear in your corpus of papers? Defining "online search" was not the problem I had--I was curious how it differed and, if it is just one method of information retrieval, why it needs to be mentioned separately.

---

> ### Comment · Reviewer_HGJ3 · 2025-09-01
>
> I agree with reviewer Kczb

---

> > ### Author Response · Authors · 2025-09-02
> >
> > Thank you all for your feedback and for expressing your concerns. We acknowledge that in our enthusiasm to share our preliminary results during the discussion period, our description of the screening process was not as clear or detailed as it should have been.
> >
> > We have developed a practical action plan, and we believe the revision is more manageable than it appears. Our core contribution (the taxonomy) has proven robust. The remaining task is a one-time, systematic enrichment and clarification of this framework, not a fundamental redesign.
> >
> > **1. Regarding the preliminary added paper screening**
> >
> > For the screening process, the ">500" represents the initial number from our search queries. We then performed a first-pass filter manually. This involved screening **titles and abstracts only** to **quickly exclude clearly irrelevant papers** and perform an **initial classification** based on them. We are currently conducting a double-check of this entire process. This process will be made clearer in the revision.
> >
> > **2. Regarding the taxonomy and integration of new papers**
> >
> > We completely agree that rigor and comprehensiveness are critical.
> >
> > We would like to clarify that **our taxonomy was designed with extensibility in mind,** which is why the new papers could be integrated so efficiently. The **taxonomy is built upon a foundational and exhaustive classification** (i.e., reference-based vs. reference-free methods), which covers existing approaches. The subsequent sections, like "LLM-based" and "domain-specific" evaluations, were then developed to provide deeper detail and structure to the rapidly growing literature within these foundational categories.
> >
> > This structure is not about rigidly forcing papers into established boxes, rather, it provides a flexible yet principled system for organizing **a rapidly evolving field**. The necessary revisions are therefore focused on: (1) systematically integrating the new literature into this robust framework, and (2) expanding the discussion section to reflect these additions. These are changes we are prepared to make now.
> >
> > **3. Regarding "Information Retrieval vs. Online Search":**
> >
> > “Information retrieval” can be used ambiguously in the literature, sometimes broadly referring to any task of querying a document for an answer (as seen in AlignScore). To address this ambiguity, we have taken care in our current revision to use this term more precisely. "online search" here specifically involves frameworks that query live search engines for dynamic, real-time web results to deal with real-time facts / time-changing facts.
> >
> > Moreover, as stated in our previous response, we want to highlight RAG systems within our 'reference-based' category based on the fact that, its retrieval component can use either a fixed knowledge base or by dynamic Online Search.
> >
> > ---
> >
> > We are fully committed to this revision and to delivering a manuscript that meets the journal's standards. We believe this plan provides a credible path to producing a methodologically sound and valuable survey for the community. We hope to have the opportunity to deliver on it.
> >
> > Thank you again to all reviewers for your feedback, helping to strengthen this paper.
> >
> > Best regards,
> >
> > Author 5135

---

### Review · Reviewer_HGJ3 · 2025-08-15

**Summary Of Contributions:**

Please see my comment, that was supposed to be my review (I must have pressed wrong button)

**Audience:**

Yes

**Broader Impact Concerns:**

Please see my comment

**Claims And Evidence:**

No

**Requested Changes:**

Please see my comment

**Strengths And Weaknesses:**

Please see my comment

---

### Comment · Reviewer_HGJ3 · 2025-08-07

I struggled with this paper because it claims to provide a “a comprehensive survey of Automatic Hallucination Evaluation (AHE) methods” but the survey is ad-hoc.  Ie, the authors looked at papers they were aware of, they did not do a structured literature survey where specific databases/repositories were searched for papers under clear inclusion/exclusion criteria (like PRISMA in medicine).  An ad-hoc survey cannot claim to be comprehensive (it is also not replicable), and indeed from my perspective their view of hallucination is very limited.

For example, I work on hallucination in medical NLG, and we do things like
•	Classify severity of hallucinations (eg “no impact on patient outcome” vs “potentially major impact on patient outcome”).  The former are undesirable but perhaps could be tolerated if infrequent; the latter are unacceptable.
•	Include as hallucinations statements which are literally true but in context likely to mislead the reader.  If reading a text could mislead a doctor and damage patient care, this is a hallucination even if the statement is literally true.
•	Acknowledge that in some cases hallucination is borderline/debateable.  For example, if facts Y in the input mean that fact X is 99% likely to be true, is it a hallucination to include X in the summary?

None of these complexities are mentioned anywhere in this paper (including Discussion and Future Directions), which reinforces my doubts as to whether the survey is “comprehensive”.   Nor do I see any mention of hallucination detection in data-to-text (eg, https://aclanthology.org/2021.inlg-1.23/), which is another area I work in.

Statements such as ““Large-scale automation in evidence gathering is thus pivotal to advancing AHE in real-world applications" are also annoying because I see no evidence that the authors are seriously engaging with hallucination detection in real-world applications.  If they were, they would highlight complexities such as above, which occur in any real-world domain (not just medicine).

So what we are left with is a summary and analysis of an ad-hoc set of papers, and an architecture (Fig 2) which fits some (not all) of the surveyed papers.

In summary, this paper would be stronger and more valuable to the research community if it was based on a proper PRISMA-like systematic literature review, and at least acknowledged the complexities of hallucinations in real-world domains.

With regard to TMLR's criteria
Summary of contributions: review and synthesis of an ad-hoc set of papers about detecting hallucinations.  Biggest weakness is that surveyed papers were selected ad-hoc, not via a structured survey

Are the claims made in the submission supported by accurate, convincing and clear evidence: Claims that the paper is a comprehensive survey of field are dubious, because survey is ad-hoc

Would at least some individuals in TMLR's audience be interested in knowing the findings of this paper: Not really valuable to me because view of hallucination is very limited, but could be valuable to people who work with some of the datasets cited.

Requested changes
* perform a proper PRISMA-like structured survey (essential)
* acknowledge complexities of real-world hallucination and how they impact hallucination detection; this could be part of discussion (essential)

Broader impact concerns: none

---

> ### Comment · Reviewer_HGJ3 · 2025-08-20
> **Clarification about scope of a survey**
>
> Let me try to clarify my comments about scope.  If a survey is a replicable scientific contribution, then it must be over a well-defined set of papers.  This could be
> * Survey of papers by one author, or one research lab
> * Survey of papers addressing a shared task, or set of shared tasks
> * Survey of papers published in specific venues, with inclusion/exclusion criteria
> * Survey of papers indexed in specific databases, with inclusion/exclusion criteria
>
> If I see a claim that a survey is a "comprehensive survey", then I generally assume that what is meant is the last, a survey over databases which are likely to include most relevant work, such as Scopus (which includes TMLR) and ACL Anthology.  Such a survey should be described using PRISMA methodology (I see another reviewer gave links for PRISMA).
>
> Unfortunately this paper does not seem to be such a survey.  The choice of papers seems ad-hoc (and certainly omits relevant papers which I am aware of).  A survey of papers which the authors have personally read or are aware of is not a replicable scientific endeavour.

---

> > ### Author Response · Authors · 2025-08-25
> >
> > Dear Reviewer HGJ3,
> >
> > Thank you for your comments and insightful feedback on our manuscript! We are grateful for your review.
> >
> > In response to your feedback, we have performed a revision that goes beyond minor edits and reframes the paper to address these core issues. We believe that the revised manuscript is now a more rigorous, comprehensive, and valuable contribution to the field.
> >
> > Below, we address each of your comments in detail.
> >
> > ---
> >
> > > It claims to provide a ‘a comprehensive survey of Automatic Hallucination Evaluation (AHE) methods’ but the survey is ad-hoc...”
> > >
> >
> > **Response:**
> > Thank you for this crucial feedback. We have taken your advice to heart and have restructured our entire literature review to follow the PRISMA framework.
> >
> > - In **Section 1.1** (Scope of the Survey), we now explicitly detail our systematic search process, including the databases queried (DBLP, ACL Anthology…), the specific keywords used, the time frame of the review, and our formal inclusion and exclusion criteria. (We selected the DBLP Computer Science Bibliography as our primary database due to its specialized focus, which ensures high relevance, comprehensive coverage of key peer-reviewed venues in AI, and structured metadata essential for a rigorous and reproducible PRISMA search.)
> > - We have also added PRISMA flow diagram (Figure 4) in Appendix A, illustrating the literature search and screening process.
> >
> > ---
> >
> > > - I work on hallucination in medical NLG, and we do things like • Classify severity of hallucinations... • Include as hallucinations statements which are literally true but in context likely to mislead the reader... • Acknowledge that in some cases hallucination is borderline/debateable... None of these complexities are mentioned anywhere in this paper... which reinforces my doubts as to whether the survey is ‘comprehensive’.”
> > > - Statements such as ‘Large-scale automation in evidence gathering is thus pivotal to advancing AHE in real-world applications" are also annoying…
> > >
> >
> > **Response:**
> > You have highlighted a critical dimension of hallucination that our initial draft did not conduct deep discussion; it should be an extension of the previous Section 2.3 (Frontiers - Domain-specific). To address the nuanced and high-stakes nature of errors in real-world applications, we have substantially revised the paper to incorporate and emphasize these complexities.
> >
> > - In the revised dataset and methodology sections (Section 2.3 and 3.4), we not only add descriptions related to medicine and law, but also expand the applications around role-play and code.
> > - We have introduced a new discussion in **Section 4.3 ("The Complexities of Hallucination in Real-World Applications")** that is dedicated to this topic.
> > - We specifically discuss the unique challenges in the medical and legal domains, and address the issue of literally true but misleading statements and the borderline/debatable nature of hallucinations within this new section, arguing that the line between an error.
> >
> > ---
> >
> > > Nor do I see any mention of hallucination detection in data-to-text...
> > >
> >
> > **Response:**
> >
> > Thank you for pointing this out. We agree that our initial draft overlooked the specific challenges of data-to-text generation, and we have revised the manuscript to properly incorporate it. While some methods we surveyed are applicable to this domain (e.g. Redeep), we did not make that connection clear.
> >
> > - We have added data-to-text generation to the **Introduction** as an application area where hallucination detection is critical.
> > - We have introduced a specific discussion in **Section 3.1.1 & 3.1.2** that explains how to retrieve table evidence and how NLI-based methods are adapted for data-to-text tasks.
> >
> > ---
> >
> > In summary, your feedback is very constructive for our paper.
> >
> > We have revised the paper towards a PRISMA-based survey and have incorporated a critical discussion on the complexities of real-world hallucinations. We are very grateful for the opportunity to revise our work and are believe that it is now a stronger and more valuable contribution to the TMLR audience.
> >
> > Thank you again for your time and guidance!
> >
> > Best regards,
> >
> > Authors of Paper5135

---

> > > ### Comment · Reviewer_HGJ3 · 2025-08-28
> > >
> > > Hi, I do not understand what you did in your PRISMA search.  My understanding is that you looked for papers that contained at least one of (hallucination, factuality, faithfulness) and one of (evaluation, assessment, measurement, benchmark) in the DBs (DBLP, ACL Anthology, Google Scholar), and this resulted in either 223+165-58 = 330 papers (Appendix A) or 165 papers (FIg 4).  The inconsistency is worrying, but what is even more worrying is that when I do a title+abstract search on ACL Anthology for papers that contain "Evaluation" and one of (hallucination, factuality, faithfulness) , I get 481 papers, which is more than what you report for your entire search.
> > >
> > > Please explain how you did the search, and whether you found 330 or 165 papers, and how this compares to the 481 papers I found above for a much more limited search.
> > >
> > > On a different topic, when discussing complexities of real-world hallucination, I suggest you look at https://arxiv.org/abs/2507.11508v1 (which will appear in EMNLP Findings) (I am not an author of this paper, by the way).  I think it does a nice job of mentioning complexities in a real-world tourism domain, including (A) categorising errors by severity of business impact, (B) counting as errors statements which are true but contextually misleading, and (C) trying to assess whether possibly subjective judgements can be considered to be errors.  The tourism domain also shows that these problems are not restricted to high-stakes domains such as medicine.
> > >
> > > Anyways, please respond about the PRISMA issue (which is more important) and explain what you did

---

> > > > ### Author Response · Authors · 2025-08-28
> > > >
> > > > Thank you for your follow-up and for the detailed query regarding our search methodology. We appreciate the opportunity to clarify our process.
> > > >
> > > > Our paper selection process combines a systematic database search (DBLP, Figure 4 Left) with a review of top-tier conference anthology (ACL Anthology and other AI conferences, Figure 4 Right) to ensure both broad coverage and high relevance.
> > > >
> > > > **Number Explained**
> > > >
> > > > 1. Systematic Database Search (via DBLP):
> > > >
> > > > - 223: means our keyword queries (e.g., "hallucination assessment", "factuality evaluation", ...) yielded a cumulative total of 223 records.
> > > > - 58: duplicate entries in DBLP (e.g., arXiv preprint and its corresponding published version) were removed.
> > > > - 165: unique entries remained (= 223 – 58).
> > > > - 27: are the duplicates between DBLP and Conference Anthology (see below).
> > > > - 138: records (= 165 – 27) were retained for screening in the next stage of the PRISMA workflow.
> > > >
> > > > 2. Systematic Conference Anthology Search ( including ACL, EMNLP, EACL, NAACL, AACL, CoNLL, CL, ICLR, AAAI, NeurIPS, ICML):
> > > >
> > > > - 106: papers were retrieved from ACL Anthology.
> > > > - 24: additional papers were retrieved from other non-ACL conference anthologies.
> > > > - 17: cited papers (from the above 106 + 24 = 130) were included, though not directly indexed in the anthology.
> > > > - 147: papers in total (106 + 24 + 17) were retained.
> > > >
> > > >
> > > > **Inconsistency with Your Search**
> > > >
> > > > It should be noted that, a keyword search on the ACL Anthology platform relies on a full-text Google search (e.g., this search link https://aclanthology.org/search/?q=hallucination+evaluation) according to their own documentation: “Note that this will still find any string that appears on such a page; it is not currently possible to search only within certain fields, such as author names or venue names.”.
> > > >
> > > > To avoid this noise from full-text search and ensure we captured papers whose primary contribution was relevant, we chose to directly review the proceedings of these core venues (through links, e.g., https://aclanthology.org/events/acl-2024/.) This allowed us to apply our inclusion criteria more accurately.
> > > >
> > > > For a paper to be included, its primary contribution should be a novel method or benchmark for Automatic Hallucination Evaluation in NLG. We excluded papers that focus merely on hallucination mitigation or multimodality. This filtering is essential for providing a focused, high-quality synthesis of AHE methodologies.
> > > >
> > > > We believe that this hybrid methodology, combining a broad systematic search with a precise, targeted venue review, provides a reliable foundation for our survey.
> > > > To ensure our work is as comprehensive as possible, we would be grateful if you could kindly share the specific link that generated the 481 results. Our understanding was that the ACL Anthology search is primarily full-text, so this would be helpful for our final cross-validation. It would allow us to perform a direct comparison and ensure no relevant category of work was inadvertently overlooked in our focused review.
> > > >
> > > > ---
> > > >
> > > > Regarding the coming EMNLP findings paper you provided, thank you for pointing us to this insightful paper. We found it highly relevant and it can help us enrich our discussion.
> > > >
> > > > We agree that its findings in the tourism domain are an excellent real-world example. The idea that different levels of hallucination carry different business impacts aligns perfectly with some of the perspectives in our paper. For instance, as we touch upon in our discussion in Section 4.3 ("The Ambiguous Boundary: Hallucination, Abstraction, and Imagination") of “factual hallucination”, which means some factually true but contextually inconsistent statements can be acceptable, or even harmless, in summarization tasks.
> > > >
> > > > To make this connection more explicit and to fully address “hallucination severity/impact”, we will add a new paragraph to Section 4.3, which will be dedicated to discussing the severity and real-world impact of hallucinations.
> > > >
> > > > ---
> > > >
> > > > Thank you again for your invaluable guidance in helping us clarify our process.
> > > >
> > > > Best,
> > > >
> > > > Authors of Paper5135

---

> > > > > ### Comment · Reviewer_HGJ3 · 2025-08-29
> > > > >
> > > > > I did my search as follows
> > > > >
> > > > > * Downloaded ACL Anthology bib file with abstracts (https://aclanthology.org/anthology+abstracts.bib.gz) (actually I used a version which I downloaded a few months ago, of course would be better to use up-to-date version)
> > > > > * Loaded this into Zotero (https://www.zotero.org/)
> > > > > * searched for bib file entries (title, abstract, also conference/workshop title) which included  "Evaluation" AND at least one of ("hallucination", "factuality", "faithfulness")
> > > > >
> > > > > This is a boolean keyword search of information in the bib file entry, its not a full text search.  Of course there are other tools which can be use (eg see https://library-guides.ucl.ac.uk/systematic-reviews/software), and indeed some people prefer to just write Python scripts which count the number of bib file entries which meet the specified criteria.
> > > > >
> > > > > Did you just search titles, rather than title+abstract?  The usual practice is to search title+abstract for keywords
> > > > >
> > > > > If you just checked papers in specific proceedings, this is not a search of the ACL Anthology, and should not be described as such.
> > > > >
> > > > > Two general points
> > > > > * you need to describe your search in sufficient detail that it can be replicated
> > > > > * a title+abstract search of the complete ACL Anthology would be better than a title-only search of selected venues within the Anthology

---

> > > > > > ### Author Response · Authors · 2025-08-29
> > > > > >
> > > > > > Thanks for providing the detailed guidance. Your instructions provided a clear path forward, and we have followed them precisely to strengthen the paper list of our survey.
> > > > > >
> > > > > > **Search Methodology**
> > > > > >
> > > > > > We have now followed your instruction on the ACL Anthology search. Specifically:
> > > > > >
> > > > > > 1. We wrote a Python script to parse the latest ACL Anthology bib file with abstracts.
> > > > > > 2. This script performed a boolean keyword search across both the **title and abstract** fields for all entries.
> > > > > > 3. The search query covered the keywords `["hallucination", "factuality", "faithfulness"]` cross-referenced with `["evaluation", "assessment", "measurement", "benchmark", "dataset"]`.
> > > > > > 4. Initial filtering was applied to exclude papers focused solely on multimodality or hallucination mitigation without proposing a new evaluation method.
> > > > > >
> > > > > > This search yielded 592 papers. After deduplicating these against our existing collection (using DOI and title), we identified 533 papers to be analyzed. We are, of course, happy to provide our Python script and the full list of filtered papers if you would find it helpful.
> > > > > >
> > > > > > Finally, we screened these 533 papers for eligibility based on our inclusion criteria. This led to the exclusion of 79 articles whose primary contribution was not the development of an AHE dataset or method, resulting in a final set of 454 relevant papers to be synthesized in our survey. We will describe this process clearly in our survey.
> > > > > >
> > > > > > **Validation of Our Proposed Taxonomy**
> > > > > >
> > > > > > Our primary goal with this survey is to synthesize and categorize the field, identifying key trends and directions. To that end, our author team has conducted a preliminary analysis of the 454 papers. The primary goal of this initial pass was to validate the structural integrity of our taxonomy, and we found that it is indeed robust enough to categorize the vast majority of these new works.
> > > > > >
> > > > > > The distribution of these papers into our proposed categories is as follows:
> > > > > > | **Main Category** | **Sub Category** | **Specific Type** | **Count** | **Percentage** |
> > > > > > | --- | --- | --- | --- | --- |
> > > > > > | **AHE Datasets and Benchmarks** |  |  | **145** | **31.94%** |
> > > > > > |  | Task-specific |  | 49 | 10.79% |
> > > > > > |  |  | Dialogue | 14 | 3.08% |
> > > > > > |  |  | Simplification | 3 | 0.66% |
> > > > > > |  |  | Summarization | 32 | 7.05% |
> > > > > > |  | General Factuality |  | 51 | 11.23% |
> > > > > > |  |  | Fact Reasoning | 3 | 0.66% |
> > > > > > |  |  | Probe Truthfulness | 27 | 5.95% |
> > > > > > |  |  | Fresh Fact | 3 | 0.66% |
> > > > > > |  |  | Knowledge-grounded QA | 18 | 3.96% |
> > > > > > |  | Application |  | 30 | 6.61% |
> > > > > > |  |  | Long Text | 10 | 2.20% |
> > > > > > |  |  | Medicine&Law | 8 | 1.76% |
> > > > > > |  |  | Non-English | 6 | 1.32% |
> > > > > > |  |  | Code | 1 | 0.22% |
> > > > > > |  |  | Other Domains | 5 | 1.10% |
> > > > > > |  | Automated Dataset Generation |  | 7 | 1.54% |
> > > > > > |  | Meta-evaluation |  | 8 | 1.76% |
> > > > > > | **AHE Methods** |  |  | **309** | **68.06%** |
> > > > > > |  | LLM-based Evaluation |  | 29 | 6.39% |
> > > > > > |  |  | Integrated Frameworks | 10 | 2.20% |
> > > > > > |  |  | LLM as a Judge | 16 | 3.52% |
> > > > > > |  |  | LLMs Cross-checking | 3 | 0.66% |
> > > > > > |  | Reference-based Evaluation |  | 204 | 44.93% |
> > > > > > |  |  | SF Evidence | 90 | 19.82% |
> > > > > > |  |  | WF Evidence | 45 | 9.91% |
> > > > > > |  |  | Lexicon-based Metrics | 3 | 0.66% |
> > > > > > |  |  | Semantics-based Metrics | 66 | 14.54% |
> > > > > > |  | Reference-free Evaluation |  | 22 | 4.85% |
> > > > > > |  |  | Consistency as a Proxy | 5 | 1.10% |
> > > > > > |  |  | Uncertainty as a Proxy | 17 | 3.74% |
> > > > > > |  | Domain-Specific AHE Frameworks |  | 54 | 11.89% |
> > > > > > |  |  | Behavioral and Systemic Consistency | 27 | 5.95% |
> > > > > > |  |  | High-Stakes Domains | 27 | 5.95% |
> > > > > > | **Total** |  |  | **454** | **100.00%** |

---

> > > > > > > ### Author Response · Authors · 2025-08-29
> > > > > > >
> > > > > > > We are double-checking this categorization for accuracy. This result validates that our initial work, while focused on high-impact papers from top-tier venues, can capture the core intellectual structure of the field.
> > > > > > >
> > > > > > > While this next stage involves incorporating new citations, it's crucial to note that this is an act of **enrichment, not a structural revision.** This demonstrates that the primary contribution of our survey, its categorization and synthesis of the field, is robust and was not skewed by the scope of our initial search. Our work is therefore to populate this validated framework with a more comprehensive evidence base.
> > > > > > >
> > > > > > > **Refinements:**
> > > > > > >
> > > > > > > The technical instruction you provided on the ACL Anthology search allows us to make the survey even more complete. Based on our analysis, we will enhance the manuscript with the following targeted additions:
> > > > > > >
> > > > > > > 1. **Hallucination in RAG Systems, in Section 3.1**
> > > > > > > While our original survey already identified RAG as a key area, this new search has revealed some other recent works (a significant part of *SF evidence* and *WF evidence* in the above table). Therefore, we will split our discussion of RAG into its own subsection. This new section will delve into the unique challenges of evaluating hallucinations in RAG, where errors can arise from multiple sources: faulty retrieval by the retriever, a lack of faithfulness by the generator, or conflicts between retrieved and parametric knowledge. It will now cover the growing body of specialized benchmarks and evaluation frameworks designed specifically for this complex interplay between retrieval and generation, an area that goes beyond standard faithfulness or factuality checks.
> > > > > > > 2. **Discussions: Broader Implications of Hallucination, in Section 4**
> > > > > > > Building on our existing discussion of the boundary between hallucination and creativity (section 4.3), the new papers allow us to add critical analysis. We will explore more on how hallucinations carry significant societal impacts (like you mentioned, the “business impact”). This could offer a more forward-thinking perspective on the broader implications of AHE research.
> > > > > > >
> > > > > > > We believe that these targeted additions, built upon the new and rigorous literature search you guided us to perform, will finalize this paper as a valuable and comprehensive resource for the community. Thank you again for your instrumental feedback.

---

> > > > > > > ### Comment · Reviewer_Kczb · 2025-08-31
> > > > > > > **How did you screen these papers?**
> > > > > > >
> > > > > > > I've been refraining from commenting while observing the updates in discussion with reviewer HGJ3, but I find it concerning that you claim to have screened >500 papers in about 5 hours. Based on my own experience conducting systematic reviews, this seems highly improbable: that is more than one paper a minute, so you would realistically need to be a group of tens of authors in order to review these papers in that time frame. This suggests that the review must be automated, which dramatically undermines the validity of the analysis (unless you have some corroborating data to demonstrate the accuracy of this automated screening process). For example, this recent paper demonstrates that "state of the art" LLMs are not adequate for systemic reviews in medicine: https://www.jmir.org/2024/1/e53164

---

### Author Response · Authors · 2025-08-25

Dear Editor and Reviewers,

Thank you all for your thorough and insightful feedback on our manuscript. We are sincerely grateful for your time and for providing such constructive comments, which have been instrumental in guiding a substantial revision of our work.

We have taken your suggestions to heart and have worked diligently to transform the paper into a more rigorous, systematic, and analytically deep survey. The most significant change, as requested, was to revise to a structured, database-driven literature survey based on PRISMA workflow.

We have conducted a structured literature search on DBLP database for papers from the last five years. This has expanded the scope of our work, leading to the inclusion of **31 new AHE methods** (for a new total of 105) and **20 new datasets** (for a new total of 70). These additions have been integrated throughout the survey's main text and tables.

We would also like to clarify that our initial search was primarily focused on the ACL Anthology and top-tier AI conference proceedings. We acknowledge that this approach may have led to some omissions, and we are deeply grateful to the reviewers for guiding us toward this more rigorous, database-centric methodology.

Guided by your suggestions and the insights from the newly included literature, we have undertaken a revision of the entire manuscript. For your convenience in reviewing the changes, we have highlighted **new content in pink** and **revisions to phrasing/typos in yellow**. For visual clarity, when a paragraph’s title is highlighted, it indicates that the entire section has been revised or added (e.g., the new paragraph "The Changing Nature of Hallucinations" in Section 4.2).

**The most important changes include:**

- **Decoupling of Datasets and Methodologies:** We no longer present datasets and AHE methodologies as a single, sequential pipeline. While our previous structure was motivated by the fact that a significant portion of evaluators (78.6%) introduce a dataset as a foundation for their method, we agree with *Reviewer tJyT* that a parallel discussion is much clearer and more logical.
- **Streamlined and Re-categorized Methodology Section:** The content previously split into "evidence collection" and "comparison" has been unified and restructured within **Section 3 (A Taxonomy of AHE Methodologies)**. This section now categorizes methods based on their core operational principles, and we have condensed overly detailed paper descriptions to improve readability and focus on overarching trends.
- **Expansion of Content:** Based on your feedback and the new literature, we have filled several content gaps:
    - **Inclusion of Data-to-Text Evaluation:** We expand hallucination evaluation in the data-to-text domain, with discussions added to the **Introduction (Section 1)** and the methodology analysis in **Section 3.1.1 & 3.1.2**.
    - **Broader Coverage of Real-World Applications:** We have greatly expanded beyond medicine and law. **Sections 2.3 and 3.4** now explore the unique challenges of hallucination detection in diverse applications such as **code generation, role-playing agents, and multi-agent systems**.
- **More In-Depth Discussion Section:** Perhaps the most notable change is the revision of **Section 4 (Discussion)**. It is now structured to address key questions about the field:
    - **Section 4.1** analyzes AHE methods through the lens of their core features and dimensions (e.g., supervision, granularity, human alignment).
    - **Section 4.2** explores the evolution of the field, comparing and contrasting past and present forms of hallucination and evaluation methods.
    - **Section 4.3** offers a more nuanced discussion on the complexities of real-world hallucination evaluation, directly addressing *Reviewer HGJ3* concerns.

We are hoping that these revisions have transformed the manuscript into a systematic and comprehensive survey. We thank you once again for your invaluable guidance, which has enabled us to produce a much stronger paper.

Sincerely,

Authors of Paper5135

---

### Note · Authors · 2025-10-13

I have read and agree with the venue's withdrawal policy on behalf of myself and my co-authors.